# Multi-task Additive Models for Robust Estimation and Automatic Structure Discovery

**Yingjie Wang**[1], **Hong Chen**[2]*, **Feng Zheng**[3], **Chen Xu**[4], **Tieliang Gong**[4,5], **Yanhong Chen**[6]

[1]College of Informatics, Huazhong Agricultural University, China
[2] College of Science, Huazhong Agricultural University, China
[3] Department of Computer Science and Engineering,
Southern University of Science and Technology, China
[4] Department of Mathematics and Statistics, University of Ottawa, Canada
[5] School of Computer Science and Technology, Xi'an Jiaotong University, China
[6] National Space Science Center, Chinese Academy of Sciences, China

## Abstract

Additive models have attracted much attention for high-dimensional regression estimation and variable selection. However, the existing models are usually limited to the single-task learning framework under the mean squared error (MSE) criterion, where the utilization of variable structure depends heavily on a priori knowledge among variables. For high-dimensional observations in real environment, *e.g.*, Coronal Mass Ejections (CMEs) data, the learning performance of previous methods may be degraded seriously due to the complex non-Gaussian noise and the insufficiency of a prior knowledge on variable structure. To tackle this problem, we propose a new class of additive models, called Multi-task Additive Models (MAM), by integrating the mode-induced metric, the structure-based regularizer, and additive hypothesis spaces into a bilevel optimization framework. Our approach does not require any priori knowledge of variable structure and suits for high-dimensional data with complex noise, *e.g.*, skewed noise, heavy-tailed noise, and outliers. A smooth iterative optimization algorithm with convergence guarantees is provided to implement MAM efficiently. Experiments on simulations and the CMEs analysis demonstrate the competitive performance of our approach for robust estimation and automatic structure discovery.

## 1 Introduction

Additive models [14], as nonparametric extension of linear models, have been extensively investigated in machine learning literatures [1, 5, 34, 44]. The attractive properties of additive models include the flexibility on function representation, the interpretability on prediction result, and the ability to circumvent the curse of dimensionality. Typical additive models are usually formulated under Tikhonov regularization schemes and fall into two categories: one focuses on recognizing dominant variables without considering the interaction among the variables [21, 28, 29, 46] and the other aims to screen informative variables at the group level, e.g., groupwise additive models [4, 42].

Although these existing models have shown promising performance, most of them are limited to the single-task learning framework under the mean squared error (MSE) criterion. Particularly, the groupwise additive models depend heavily on a priori knowledge of variable structure. In this paper, we consider a problem commonly encountered in multi-task learning, in which all tasks share an underlying variable structure and involve data with complex non-Gaussian noises, e.g., skewed

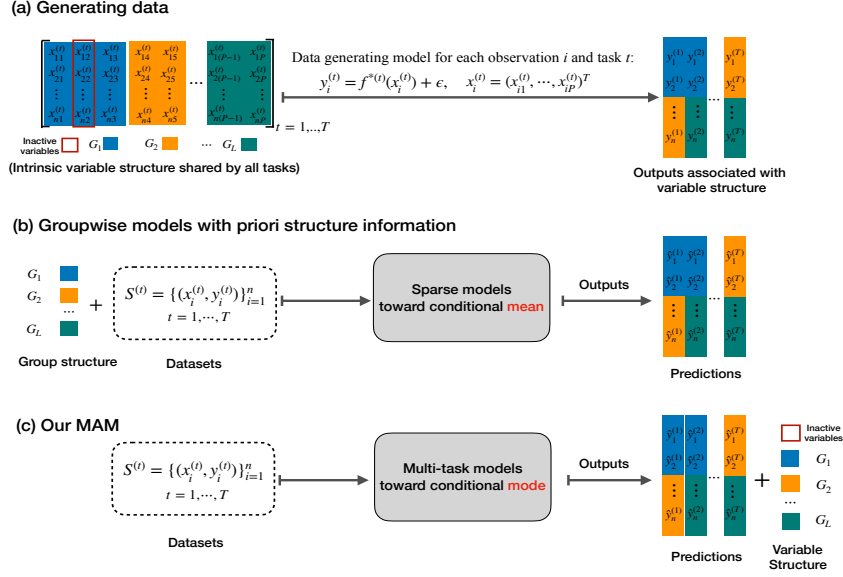

Figure 1: A schematic comparison between previous groupwise models and MAM. (a) Data generating process. (b) Mean-based groupwise models with a priori knowledge of group structure, e.g., group lasso [33, 43] and group additive models [42]. (c) Mode-based MAM for robust estimation and automatic structure discovery.

noise, heavy-tailed noise, and outliers. The main motivation of this paper is described in Figure 1. As shown in Figure 1(a), the intrinsic variable structure for generating data is encoded by several variable groups $\{G_1, G_2, ..., G_L\}$, where some groups also contain inactive variables. For each task $t \in \{1, ..., T\}$, the output is related to different dominant groups, e.g., $G_1, G_2$ for the first task. With a prior knowledge of group structure, single-task groupwise models shown in Figure 1(b) aim to estimate the conditional mean independently, e.g., group lasso [13, 22, 33, 43] and group additive models [4, 16, 42]. All above models are formulated based on a prior knowledge of group structure and Gaussian noise assumption. However, these requirements are difficult to be satisfied in real applications, e.g., Coronal Mass Ejections (CMEs) analysis [20].

To relax the dependence on a prior structure and Gaussian noise, this paper proposes a class of Multi-task Additive Models (MAM) by integrating additive hypothesis space, mode-induced metric [6, 41, 10], and structure-based regularizer [12] into a bilevel learning framework. The bilevel learning framework is a special kind of mathematical program related closely with optimization schemes in [7, 12]. A brief overview of MAM is shown in Figure 1(c). The proposed MAM can achieve robust estimation under complex noise and realize data-driven variable structure discovery. The main contributions of this paper are summarized as below:

- *Model:* A new class of multi-task additive models is formulated by bringing four distinct concepts (e.g., multi-task learning [2, 9], sparse additive models [3, 4, 18, 42], mode-induced metric [10, 38], and bilevel learning framework [12, 32]) together in a coherent way to realize robust and interpretable learning. As far as we know, these issues have not been unified in a similar fashion before.

- *Optimization:* An optimization algorithm is presented for the non-convex and non-smooth MAM by integrating Half Quadratic (HQ) optimization [24] and dual Forward-Backward algorithm with Bregman distance (DFBB) [37] into proxSAGA [30]. In theory, we provide the convergence analysis of the proposed optimization algorithm.

- *Effectiveness:* Empirical effectiveness of the proposed MAM is supported by experimental evaluations on simulated data and CMEs data. Experimental results demonstrate that MAM can identify variable structure automatically and estimate the intrinsic function efficiently even if the datasets are contaminated by non-Gaussian noise.

Table 1: Algorithmic properties (✓-has the given information, ×-hasn't the given information)

|  | RMR [38] | GroupSpAM [42] | CGSI[26] | BIGL[12] | MAM (ours) |
|---|---|---|---|---|---|
| Hypothesis Space | Linear | Additive | Additive | Linear | Additive |
| Learning Task | Single-task | Single-task | Single-task | Multi-task | Multi-task |
| Evaluation Criterion | Mode-induced | Mean-induced | Mean-induced | Mean-induced | Mode-induced |
| Objective Function | Nonconvex | Convex | Convex | Convex | Nonconvex |
| Robust Estimation | ✓ | × | × | × | ✓ |
| Sparsity on Grouped Features | × | ✓ | ✓ | ✓ | ✓ |
| Sparsity on Individual Features | ✓ | × | × | × | ✓ |
| Variable Structure Discovery | × | × | ✓ | ✓ | ✓ |

**Related works**: There are some works for automatic structure discovery in additive models [26, 40] and partially linear models [19, 45]. Different from our MAM, these approaches are formulated under single-task framework and the MSE criterion, which are sensitive to non-Gaussian noise and difficult to tackle multi-task structure discovery directly. While some mode-based approaches have been designed for robust estimation, e.g., regularized modal regression (RMR) [38], none of them consider the automatic structure discovery. Recently, an extension of group lasso is formulated for variable structure discovery [12]. Although this approach can induce the data-driven sparsity at the group level, it is limited to the linear mean regression and ignores the sparsity with respect to individual features. To better highlight the novelty of MAM, its algorithmic properties are summarized in Table 1, compared with RMR [38], Group Sparse Additive Models (GroupSpAM) [42], Capacity-based group structure identification (CGSI)[26], and Bilevel learning of Group Lasso (BiGL) [12].

## 2  Multi-task Additive Models

### 2.1  Additive models

Now recall some backgrounds of additive models [14, 42, 44]. For the sake of readability, we summarize some necessary notations in ***Supplementary Material A***.

Let $\mathcal{X} \subset \mathbb{R}^P$ be the input space and $\mathcal{Y} \subset \mathbb{R}$ be the corresponding output set. We consider the following data-generating model

$$Y = f^*(X) + \epsilon, \tag{1}$$

where $X \in \mathcal{X}, Y \in \mathcal{Y}$, $\epsilon$ is a random noise, and $f^*$ is the ground truth function. For simplicity, denote $\rho(X, Y)$ as the intrinsic distribution generated in (1). Under the Gaussian noise assumption, i.e. $\mathbb{E}(\epsilon|X = x) = 0$, a large family of nonparametric regression aims to estimate the conditional mean function $f^*(x) = \mathbb{E}(Y|X = x)$. However, the nonparametric regression may face low convergence rate due to the so-called curse of dimensionality [18, 34]. This motivates the research on additive models [14, 29] to remedy this problem.

*Additive Models* [14, 29]: Let the input space $\mathcal{X} = (\mathcal{X}_1, ..., \mathcal{X}_P)^T \subset \mathbb{R}^P$ and let the hypothesis space with additive structure be defined as

$$\mathcal{H} = \left\{ f : f(\mathbf{u}) = \sum_{j=1}^{P} f_j(u_j), f_j \in \mathcal{H}_j, \mathbf{u} = (u_1, ..., u_P)^T, u_j \in \mathcal{X}_j \right\},$$

where $\mathcal{H}_j$ is the component function space on $\mathcal{X}_j$. Usually, additive models aim to find the minimizer of $\mathbb{E}(Y - f(X))^2$ in $\mathcal{H}$. Moreover, groupwise additive models have been proposed with the help of a prior knowledge of variable group, e.g., GroupSpAM [42] and GroupSAM [4].

Let $\{G_1, G_2, ..., G_L\}$ be a partition over variable indices $\{1, ..., P\}$ such that $G_l \cap G_j = \emptyset, \forall l \neq j$ and $\cup_{l=1}^{L} G_l = \{1, ..., P\}$. In essential, the main purpose of GroupSpAM [42] is to search the minimizer of

$$\mathbb{E}(Y - f(X))^2 + \sum_{l=1}^{L} \tau_l \sqrt{\sum_{j \in G_l} \mathbb{E}[f_j^2(u_j)]} \text{ over all } f = \sum_{l=1}^{L} \sum_{j \in G_l} f_j \in \mathcal{H},$$

where $\tau_l$ is the corresponding weight for group $G_l, 1 \leq l \leq L$.

### 2.2  Mode-induced metric

Beyond the Gaussian noise assumption in [16, 29, 42], we impose a weaker assumption on $\epsilon$, i.e., $\arg\max_{t \in \mathbb{R}} p_{\epsilon|X}(t) = 0$, where $p_{\epsilon|X}$ denotes the conditional density function of $\epsilon$ given $X$. In

theory, this zero-mode assumption allows for more complex cases, *e.g.*, Gaussian noise, heavy-tailed noise, skewed noise or outliers.

Denote $p(Y|X = x)$ as the conditional density function of $Y$ given $X = x$. By taking mode on the both sides of (1), we obtain the conditional mode function

$$f^*(x) = \arg\max_{t \in \mathbb{R}} p(t|X = x), \tag{2}$$

where $\arg\max_{t \in \mathbb{R}} p(t|X = x)$ is assumed to be unique for any $x \in \mathcal{X}$. There are direct strategy and indirect strategy for estimating $f^*$ [31]. Generally, the direct approaches are intractable since the conditional mode function cannot be elicited directly [15], while the indirect estimators based on kernel density estimation (KDE) have shown promising performance [6, 10, 38, 41].

Now, we introduce a mode-induced metric [10, 38] associated with KDE. For any measurable function $f : \mathcal{X} \to \mathbb{R}$, the mode-induced metric is

$$\mathcal{R}(f) = \int_{\mathcal{X}} p_{Y|X}(f(x)|X = x)d\rho_{\mathcal{X}}(x), \tag{3}$$

where $\rho_{\mathcal{X}}$ is the marginal distribution of $\rho$ with respect to $\mathcal{X}$. As discussed in [10], $f^*$ is the maximizer of the mode-induced metric $\mathcal{R}(f)$. According to Theorem 5 in [10], we have $\mathcal{R}(f) = p_{E_f}(0)$, where $p_{E_f}(0)$ is the density function of error random variable $E_f = Y - f(X)$.

Define a modal kernel $\phi$ such that $\forall u \in \mathbb{R}, \phi(u) = \phi(-u), \phi(u) > 0$ and $\int_{\mathbb{R}} \phi(u)du = 1$. Typical examples of modal kernel include Gaussian kernel, Logistic kernel, Epanechnikov kernel. Given $\{(x_i, y_i)\}_{i=1}^n \subset \mathcal{X} \times \mathcal{Y}$, an empirical version of $\mathcal{R}(f)$ obtained via KDE [10, 27] is defined as

$$\mathcal{R}_{emp}^{\sigma}(f) = \frac{1}{n\sigma} \sum_{i=1}^n \phi\Big(\frac{y_i - f(x_i)}{\sigma}\Big), \tag{4}$$

where $\sigma$ is a positive bandwidth. Then, denote the data-free robust metric w.r.t. $\mathcal{R}_{emp}^{\sigma}(f)$ as

$$\mathcal{R}^{\sigma}(f) = \frac{1}{\sigma} \int_{\mathcal{X} \times \mathcal{Y}} \phi\Big(\frac{y - f(x)}{\sigma}\Big)d\rho(x, y). \tag{5}$$

Theorem 10 in [10] states that $\mathcal{R}^{\sigma}(f)$ tends to $\mathcal{R}(f)$ when $\sigma \to 0$.

## 2.3 Mode-induced group additive models

Here, we form the additive hypothesis space based on smoothing splines [16, 23, 29, 46]. Let $\{\psi_{jk} : k = 1, ..., \infty\}$ be bounded and orthonormal basis functions on $\mathcal{X}_j$. Then the component function space can be defined as $\bar{\mathcal{B}}_j = \Big\{\bar{f}_j : \bar{f}_j = \sum_{k=1}^{\infty} \beta_{jk}\psi_{jk}(\cdot)\Big\}$ with the coefficient $\beta_{jk}, j = 1, ..., P$. After truncating these basis functions to finite dimension $d$, we get

$$\mathcal{B}_j = \Big\{f_j : f_j = \sum_{k=1}^d \beta_{jk}\psi_{jk}(\cdot)\Big\}.$$

Denote $\|f\|_2 := \sqrt{\int f^2(x)dx}$. It has been illustrated that $\|f_j - \bar{f}_j\|_2^2 = \mathcal{O}(1/d^4)$ for the second order Sobolev ball $\bar{\mathcal{B}}_j$[46]. The mode-induced Group Additive Models (mGAM) can be formulated as

$$\hat{f} = \arg\max_{f=\sum_{j=1}^P f_j, f_j \in \mathcal{B}_j} \{\mathcal{R}_{emp}^{\sigma}(f) - \lambda\Omega(f)\}, \tag{6}$$

where $\lambda$ is a positive regularization parameter and the structure-based regularizer

$$\Omega(f) = \sum_{l=1}^L \tau_l \sqrt{\sum_{j \in G_l} \|f_j\|_2^2} = \sum_{l=1}^L \tau_l \sqrt{\sum_{j \in G_l} \sum_{k=1}^d \beta_{jk}^2}$$

with group weight $\tau_l$. Denote $\Psi_i = \big(\psi_{11}(x_{i1}), ..., \psi_{1d}(x_{i1}), ..., \psi_{P1}(x_{iP}), ..., \psi_{Pd}(x_{iP})\big)$ and $\beta = (\beta_{11}, ..., \beta_{1d}, ..., \beta_{P1}, ..., \beta_{Pd})^T \in \mathbb{R}^{Pd}$. Given observations $\{(x_i, y_i)\}_{i=1}^n$ with $x_i =$

$(x_{i1}, ..., x_{iP})^T \in \mathbb{R}^P$, the mGAM can be represented as

$$\hat{f} = \sum_{j=1}^{P} \hat{f}_j = \sum_{j=1}^{P} \sum_{k=1}^{d} \hat{\beta}_{jk} \psi_{jk}(\cdot)$$

with

$$\hat{\beta} = \arg \max_{\beta \in \mathbb{R}^{Pd}} \left\{ \frac{1}{n\sigma} \sum_{i=1}^{n} \phi\left(\frac{y_i - \Psi_i \beta}{\sigma}\right) - \lambda \sum_{l=1}^{L} \tau_l \sqrt{\sum_{j \in G_l} \sum_{k=1}^{d} \beta_{jk}^2} \right\}. \tag{7}$$

**Remark 1.** *The mGAM is a robust extension of GroupSpAM from mean regression to mode regression. When each group $G_l, l \in \{1, ..., L\}$ is a singleton, our mGAM reduces to a robust version of SpAM [29] by replacing the MSE with the robust mode-induced metric (3). In particular, our mGAM is consistent with RMR [38] when each group is a singleton and all component functions are linear.*

## 2.4 Multi-task additive models

To reduce the dependency of mGAM on a priori structure information, this section formulates MAM by learning an augmented mGAM within a multi-task bilevel framework [11, 12, 25].

Let $T$ be the number of tasks. Let $\mathcal{X}^{(t)} = (\mathcal{X}_1^{(t)}, ..., \mathcal{X}_P^{(t)})^T \subset \mathbb{R}^P$ and $\mathcal{Y}^{(t)} \subset \mathbb{R}$ be the input space and the output space respectively associated with the $t$-th task. Suppose that observations $S^{(t)} = \{x_i^{(t)}, y_i^{(t)}\}_{i=1}^{2n} \subset \mathcal{X}^{(t)} \times \mathcal{Y}^{(t)}$ are drawn from an unknown distribution $\rho^{(t)}(x, y)$. Without loss of generality, we split each $S^{(t)}$ into the training set $S_{train}^{(t)}$ and the validation set $S_{val}^{(t)}$ with the same sample size $n$ for subsequent analysis.

To quantify the groups $\{G_1, ..., G_L\}$, we introduce the following unit simplex

$$\Theta = \left\{ \vartheta = (\vartheta_1, ..., \vartheta_L) \in \mathbb{R}^{P \times L} \,\middle|\, \sum_{l=1}^{L} \vartheta_{jl} = 1, 0 \le \vartheta_{jl} \le 1, j = 1, ..., P \right\},$$

where each element $\vartheta_{jl}$ can be viewed as a probability that identifies whether the $j$-th variable belongs to group $G_l$. It is desirable to enjoy the property that $\vartheta_{jl} = 1 \Rightarrow j \in G_l$ and $\vartheta_{jl} = 0 \Rightarrow j \notin G_l$. However, we cannot mine the sparsity within each group since $\sum_{l=1}^{L} \vartheta_{jl} = 1, j = 1, ..., P$. Inspired from [35], we introduce $\nu = (\nu_1, ..., \nu_P)^T \in [0, 1]^P$ to screen main effect variables across all tasks, where $\nu_j \neq 0$ means the $j$-th variable is effective.

Denote $\Psi_i^{(t)} = (\psi_{11}(x_{i1}^{(t)}), ..., \psi_{1d}(x_{i1}^{(t)}), ..., \psi_{P1}(x_{iP}^{(t)}), ..., \psi_{Pd}(x_{iP}^{(t)}))$. Given $\{S_{val}^{(t)}\}_{t=1}^{T}$ and $\{S_{train}^{(t)}\}_{t=1}^{T}$, our MAM can be formulated as the following bilevel optimization scheme:

**Outer Problem (based on validation set $S_{val}^{(t)}$):**

$$(\hat{\vartheta}, \hat{\nu}) \in \arg\max_{\vartheta \in \Theta, \nu \in [0,1]^P} \sum_{t=1}^{T} U(\hat{\beta}^{(t)}(\vartheta), \nu) \quad \text{with} \quad U(\hat{\beta}^{(t)}(\vartheta), \nu) = \frac{1}{n\sigma} \sum_{i=1}^{n} \phi\left(\frac{y_i^{(t)} - \Psi_i^{(t)} \mathcal{T}_\nu \hat{\beta}^{(t)}(\vartheta)}{\sigma}\right),$$

where $\mathcal{T}_\nu$ is a linear operator for screening main effect variables across all tasks such that $\mathcal{T}_\nu \hat{\beta}^{(t)}(\vartheta) = (\nu_1 \hat{\beta}_{11}^{(t)}(\vartheta), ..., \nu_1 \hat{\beta}_{1d}^{(t)}(\vartheta), ..., \nu_P \hat{\beta}_{P1}^{(t)}(\vartheta), ..., \nu_P \hat{\beta}_{Pd}^{(t)}(\vartheta))^T \in \mathbb{R}^{Pd}$, and $\hat{\beta}(\vartheta) = (\hat{\beta}^{(t)}(\vartheta))_{1 \le t \le T}$ is the maximizer of the following augmented mGAM:

**Inner Problem (based on training set $S_{train}^{(t)}$):**

$$\hat{\beta}(\vartheta) = \arg\max_{\beta} \sum_{t=1}^{T} J(\beta^{(t)}) \text{ with } J(\beta^{(t)}) = \frac{1}{n\sigma} \sum_{i=1}^{n} \phi\left(\frac{y_i^{(t)} - \Psi_i^{(t)} \beta^{(t)}}{\sigma}\right) - \frac{\mu}{2} \|\beta^{(t)}\|_2^2 - \lambda \sum_{l=1}^{L} \tau_l \|\mathcal{T}_{\vartheta_l} \beta^{(t)}\|_2,$$

where $\mathcal{T}_{\vartheta_l} \beta^{(t)} = (\vartheta_{1l} \beta_{11}^{(t)}, ..., \vartheta_{1l} \beta_{1d}^{(t)}, ..., \vartheta_{Pl} \beta_{P1}^{(t)}, ..., \vartheta_{Pl} \beta_{Pd}^{(t)})^T \in \mathbb{R}^{Pd}$ is used for identifying which variables belong to the $l$-th group, and the penalty term $\frac{\mu}{2} \|\beta^{(t)}\|_2^2$ with a tending-to-zero parameter $\mu$ assures the strong-convex property for optimization.

Finally, the multi-task additive models (MAM) can be represented as below:

$$\hat{f}^{(t)} = \sum_{j=1}^{P} \sum_{k=1}^{d} \hat{\nu}_j \hat{\beta}_{jk}^{(t)}(\hat{\vartheta}) \psi_{jk}(\cdot), \ t = 1, .., T.$$

Let $\hat{\vartheta}^{\text{Thr}}$ and $\hat{\nu}^{\text{Thr}}$ be two threshold counterparts of $\hat{\vartheta}$ and $\hat{\nu}$, respectively. Similar with [12], $\hat{\vartheta}^{\text{Thr}}$ is determined by assigning each feature to its most dominant group. For any $j = 1, ..., P$, $\hat{\nu}_j^{\text{Thr}}$ is determined by a threshold $u$, i.e., $\hat{\nu}_j^{\text{Thr}} = 0$ if $\hat{\nu}_j \le u$, and $\hat{\nu}_j^{\text{Thr}} = 1$ otherwise. Then the data-driven variable structure can be obtained via $\hat{\mathcal{S}} = (\hat{\vartheta}_l^{\text{Thr}} \odot \hat{\nu}^{\text{Thr}})_{1 \le l \le L}$, where $\odot$ denotes Hadamard product.

**Remark 2.** *If the hyper-parameter $\nu \equiv \mathbb{I}_P$, the sparsity w.r.t individual features would not be taken into account for MAM. In this setting, our MAM is essentially a robust and nonlinear extension of BiGL [11] by incorporating mode-induced metric and additive hypothesis space.*

**Remark 3.** *Indeed, mGAM with an oracle variable structure is the baseline of MAM. In other words, the inner problem with the estimated variable structure $\hat{\mathcal{S}}$ aims to approximate the mGAM.*

---

**Algorithm 1:** Prox-SAGA for MAM

---

**Input**: Data $\{S_{train}^{(t)}, S_{val}^{(t)}\}_{t=1}^{T}$, Max-Iter $Z \in \mathbb{R}$, The number of groups $L$, Step-size $\eta_\vartheta$, Step-size $\eta_\nu$, $\vartheta^{(0)}$, $\nu^{(0)}$, $\lambda$, $\mu$, Modal kernel $\phi$, Bandwidth $\sigma$, Weights $\tau_l, l = 1, ..., L$.

**Initialization**: $a_t = \vartheta^{(0)}, c_t = \nu^{(0)}, t = 1, ..., T, g_\vartheta^{(0)} = 0_{P \times L}, g_\nu^{(0)} = 0_P$.

**for** $z = 0, 1, ..., Z - 1$ **do**

  1. **Randomly pick set:**
      $B^{(z)} \subset \{1, ..., T\}$, denote its cardinality as $|B^{(z)}|$.

  2. **Compute** $\hat{\beta}^{(k)}(\vartheta^{(z)})$ **based on** $S_{train}^{(k)}$, $\forall k \in B^{(z)}$:
      $\hat{\beta}^{(k)}(\vartheta^{(z)})$=HQ-DFBB$(\vartheta^{(z)}, \lambda, \sigma, \mu, \tau; S_{train}^{(k)})$.

  3. **Update** $\vartheta$ **based on** $S_{val}^{(k)}$:
     3.1): $G_\vartheta = \frac{1}{|B^{(z)}|} \sum_{k \in B^{(z)}} \left( h_\vartheta(\hat{\beta}^{(k)}(\vartheta^{(z)}), \nu^{(z)}) - h_\vartheta(\hat{\beta}^{(k)}(a_k), \nu^{(z)}) \right)$.
     3.2): $\bar{\vartheta}^{(z)} = g_\vartheta^{(z)} + G_\vartheta$.
     3.3): $\vartheta^{(z+1)} = \mathcal{P}_\vartheta(\vartheta^{(z)} - \eta_\vartheta \bar{\vartheta}^{(z)})$.
     3.4): $g_\vartheta^{(z+1)} = g_\vartheta^{(z)} + \frac{|B^{(z)}|}{T} G_\vartheta$.
     3.5): $a_k = \vartheta^{(z)}$, for every $k \in B^{(z)}$.

  4. **Update** $\nu$ **based on** $S_{val}^{(k)}$:
     4.1): $G_\nu = \frac{1}{|B^{(z)}|} \sum_{k \in B^{(z)}} \left( h_\nu(\hat{\beta}^{(k)}(\vartheta^{(z)}), \nu^{(z)}) - h_\nu(\hat{\beta}^{(k)}(\vartheta^{(z)}), c_k) \right)$.
     4.2): $\bar{\nu}^{(z)} = g_\nu^{(z)} + G_\nu$.
     4.3): $\nu^{(z+1)} = \mathcal{P}_\nu(\nu^{(z)} - \eta_\nu \bar{\nu}^{(z)})$.
     4.4): $g_\nu^{(z+1)} = g_\nu^{(z)} + \frac{|B^{(z)}|}{T} G_\nu$.
     4.5): $c_k = \nu^{(z)}$, for every $k \in B^{(z)}$.

**Output:** $\hat{\vartheta} = \vartheta^{(Z)}, \hat{\nu} = \nu^{(Z)}, \hat{\beta}^{(t)}(\hat{\vartheta}), t = 1, ..., T$;

**Prediction function:** $\hat{f}^{(t)} = \sum_{j=1}^{P} \sum_{k=1}^{d} \hat{\nu}_j \hat{\beta}_{jk}^{(t)}(\hat{\vartheta}) \psi_{jk}(\cdot), \ t = 1, ..., T$;

**Variable structure:** $\hat{\mathcal{S}} = (\hat{\vartheta}_l^{\text{Thr}} \odot \hat{\nu}^{\text{Thr}})_{1 \le l \le L}$.

---

## 3   Optimization Algorithm

To implement the non-convex and nonsmooth MAM, we employ Prox-SAGA algorithm [30] with simplex projection and box projection [8]. For simplicity, we define two partial derivative calculators:

$$-\sum_{t=1}^{T} \frac{\partial U(\hat{\beta}^{(t)}(\vartheta), \nu)}{\partial \nu} := \sum_{t=1}^{T} h_\nu(\hat{\beta}^{(t)}(\vartheta), \nu), \qquad -\sum_{t=1}^{T} \frac{\partial U(\hat{\beta}^{(t)}(\vartheta), \nu)}{\partial \vartheta} := \sum_{t=1}^{T} h_\vartheta(\hat{\beta}^{(t)}(\vartheta), \nu).$$

It is trivial to compute $\sum_{t=1}^{T} h_\nu(\hat{\beta}^{(t)}(\vartheta), \nu)$ since the parameter $\nu$ only appears explicitly in the upper problem. The optimization parameter $\vartheta$ is implicit via the solution $\hat{\beta}(\vartheta)$ of the inner problem. Hence,

computing $\sum_{t=1}^{T} h_\vartheta(\hat{\beta}^{(t)}(\vartheta), \nu)$ requires us to develop a smooth algorithm HQ-DFBB (combining HQ [24] and DFBB [37]) for the solution $\hat{\beta}(\vartheta)$. For the space limitation, the optimization details including HQ-DFBB and two partial derivative calculators are provided in ***Supplementary Material B***. Let $\mathcal{P}_\Theta$ be the projection onto unit simplex $\Theta$, and $\mathcal{P}_\nu$ be the box projection onto $[0,1]^P$. The general procedure of Prox-SAGA is summarized in Algorithm 1.

**Remark 4.** *From Theorem 2.1 in [12] and Theorem 4 in [30], we know that Algorithm 1 converges only if the iteration sequence generated by HQ-DFBB converges to the solution of the inner problem. Detailed convergence analysis of HQ-DFBB is provided in **Supplementary Material C**.*

## 4 Experiments

This section validates the effectiveness of MAM on simulated data and CMEs data. All experiments are implemented in MATLAB 2019b on an intel Core i7 with 16 GB memory.

### 4.1 Simulated data analysis

**Baselines**: The proposed MAM is compared with BiGL [11] in terms of variable structure recovery and prediction ability. In addition, we also consider some baselines, including Lasso [36], RMR [38], mGAM, Group Lasso (GL) [43] and GroupSpAM [42]. Note that the oracle variable structure is a priori knowledge for implementing mGAM, GL and GroupSpAM.

**Oracle variable structure**: Set the number of tasks $T = 500$, the dimension $P = 50$ for each task and the actual number of groups $L^* = 5$. We denote the indices of $l$-th group by $G_l = \{1 + (l-1)(P/L^*), ..., l(P/L^*)\}, \forall l \in \{1, ..., L^*\}$. In addition, we randomly pick $\mathcal{V} \subset \{1, ..., P\}$ to generate sparse features across all tasks. For each $j \in \{1, ..., P\}$ and $l \in \{1, ..., L^*\}$, the oracle variable structure $\mathcal{S}^*$ can be defined as $\mathcal{S}_{jl}^* = 1$ if $j \in \mathcal{V}^c \cap G_l$, and 0 otherwise.

**Parameter selection**: For the same hyper-parameters in BiGL and MAM, we set $Z = 3000$, $\mu = 10^{-3}$, $M = 5$, $Q = 100$ and $\sigma = 2$. We search the regularization parameter $\lambda$ in the range of $\{10^{-4}, 10^{-3}, 10^{-2}, 10^{-1}\}$. Here, we assume the actual number of groups is known, i.e., $L = L^*$. The weight for each group is set to be $\tau_l = 1, \forall l \in \{1, ..., L\}$. Following the same strategy in [11], we choose the initialization $\vartheta^{(0)} = \mathcal{P}_\vartheta(\frac{1}{L}\mathbb{I}_{P \times L} + 0.01\mathcal{N}(0_{P \times L}, \mathbb{I}_{P \times L})) \in \mathbb{R}^{P \times L}$ and $\nu^{(0)} = (0.5, ..., 0.5)^T \in \mathbb{R}^P$.

**Evaluation criterion**: Denote $\hat{f}^{(t)}$, $f^{*(t)}$ as the estimator and ground truth function respectively, $1 \le t \le T$. Evaluation criterions used here include *Average Square Error(ASE)*$=\frac{1}{T}\sum_{t=1}^{T}\frac{1}{n}\|\hat{f}^{(t)} - y^{(t)}\|_2^2$, *True Deviation (TD)*$= \frac{1}{T}\sum_{t=1}^{T}\frac{1}{n}\|\hat{f}^{(t)} - f^{*(t)}\|_2^2$, *Variable Structure Recovery* $\hat{\mathcal{S}} = (\nu^{\mathrm{Thr}} \odot \vartheta_l^{\mathrm{Thr}})_{1 \le l \le L}$ with the hard threshold value $u = 0.5$, *Width of Prediction Intervals (WPI)* and *Sample Coverage Probability (SCP)* with the confidence level $10\%$. Specially, WPI and SCP are designed in [41] for comparing the widths of the prediction intervals with the same confidence level (see Section 3.2 in [41] for more details).

**Data sets**: The training set, validation set and test set are all drawn from $y^{(t)} = f^{*(t)}(\mathbf{u}^{(t)}) + \epsilon$ with the same sample size $n = 50$ for each task, where $\mathbf{u}^{(t)} = (u_1, ..., u_P)^T \in \mathbb{R}^P$ is randomly drawn from Gaussian distribution $\mathcal{N}(0_P, \frac{1}{2}\mathbb{I}_P)$. The noise $\epsilon$ follows Gaussian noise $\mathcal{N}(0, 0.05)$, Student noise $t(2)$, Chi-square noise $\mathcal{X}^2(2)$ and Exponential noise $\mathrm{Exp}(2)$, respectively. We randomly pick $\mathcal{G}^{(t)} \subset \{G_1, ..., G_L\}$ s.t. $|\mathcal{G}^{(t)}| = 2$, and consider the following examples of ground truth function $f^{*(t)}, 1 \le t \le T$:

*Example A [12].* Linear component function $f^{*(t)}(\mathbf{u}^{(t)}) = \sum_{G_l \in \mathcal{G}^{(t)}} \sum_{j \in G_l \cap \mathcal{V}^c} u_j^{(t)} \beta_j^{(t)}$, where the true regression coefficient $\beta_j^{(t)} = 1$ if $j \in G_l \cap \mathcal{V}^c$, otherwise $\beta_j^{(t)} = 0$.

*Example B.* Denote $f_1^*(u) = 2.5\sin(u)$, $f_2^*(u) = 2u$, $f_3^*(u) = 2e^u - e^{-1} - 1$, $f_4^*(u) = 8u^2$ and $f_5^*(u) = 3\sin(2e^u)$. The nonlinear additive function $f^{*(t)}(\mathbf{u}^{(t)}) = \sum_{G_l \in \mathcal{G}^{(t)}} \sum_{j \in G_l \cap \mathcal{V}^c} f_l^*(u_j^{(t)})$.

Here, spline basis matrix for MAM, mGAM and GroupSpAM are constructed with $d = 3$. In the data-generating process, we consider two cases of the number of inactive variables, i.e., $|\mathcal{V}| = 0$ and $|\mathcal{V}| = 5$. Due to the space limitation, we only present the results with Gaussian noise and

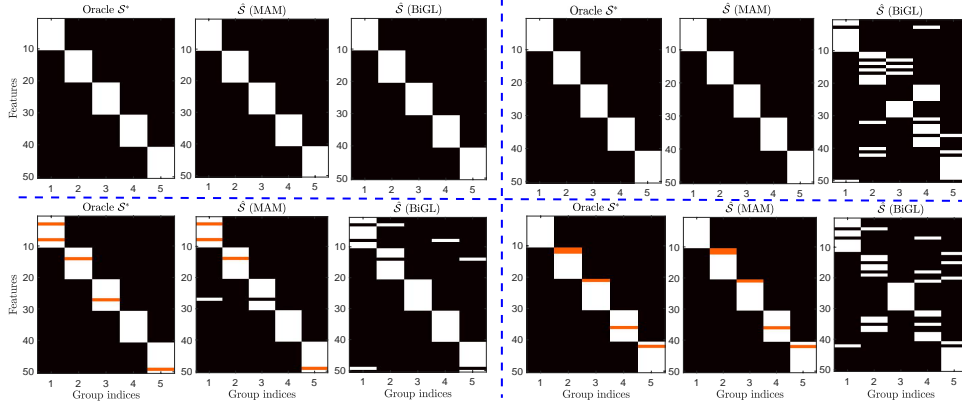

Figure 2: Variable structure discovery for *Example A* under different noise and sparsity index $\mathcal{V}$ (white pixel means the grouped variables and red pixel means the inactive variables). Top left panel: *Gaussian noise* with $|\mathcal{V}| = 0$. Top right panel: *Student noise* with $|\mathcal{V}| = 0$. Bottom left panel: *Gaussian noise* with $|\mathcal{V}| = 5$. Bottom right panel: *Student noise* with $|\mathcal{V}| = 5$.

Student noise in Table 2 and Figure 2. The remaining results, as well as several evaluations on the impact of hyper-parameters, are provided in ***Supplementary Material D.1***. From the reported results, even without the structure information, the proposed MAM can provide the competitive regression estimation with mGAM (given priori structure), and usually achieve better performance than these competitors when the noise is non-Gaussian distribution. Specially, the actual number of groups is assumed to be known in current evaluations, i.e., $L = L^*$. In ***Supplementary Material D.1***, we further verify the effectiveness of MAM for the general setting $L > L^*$.

Table 2: Performance comparisons on *Example A* (top) and *Example B* (bottom) w.r.t different criterions.

| Methods | $|\mathcal{V}| = 0$ (Gaussian noise) | | | $|\mathcal{V}| = 5$ (Gaussian noise) | | | $|\mathcal{V}| = 0$ (Student noise) | | | $|\mathcal{V}| = 5$ (Student noise) | | |
|---|---|---|---|---|---|---|---|---|---|---|---|---|
| | ASE | TD | WPI (SCP) | ASE | TD | WPI (SCP) | ASE | TD | WPI (SCP) | ASE | TD | WPI (SCP) |
| MAM | 0.0920 | 0.0914 | 0.0679(0.1015) | 0.0801 | 0.0794 | 0.0595(0.1014) | 1.0204 | 0.2467 | 0.1817(0.1014) | 1.4279 | 0.2269 | 0.1796(0.1020) |
| BiGL | 0.0715 | 0.0701 | **0.0616**(0.1014) | 0.0799 | 0.0793 | 0.0553(0.1027) | 1.1097 | 0.3550 | 0.2061(**0.1023**) | 1.4961 | 0.3544 | 0.2093(0.1017) |
| mGAM | 0.0894 | 0.0885 | 0.0651(0.1011) | 0.0795 | 0.0788 | 0.0611(0.1016) | **1.0132** | **0.2441** | **0.1803**(0.1015) | 1.3828 | 0.2242 | **0.1725**(**0.1026**) |
| GL | **0.0683** | **0.0661** | 0.0620(0.1014) | **0.0708** | **0.0684** | **0.0535**(0.1011) | 1.3252 | 0.3163 | 0.1935(0.1020) | 1.4811 | 0.3576 | 0.1976(0.1020) |
| Lasso | 0.2145 | 0.2138 | 0.1011(0.1013) | 0.2204 | 0.2131 | 0.1080(0.1017) | 3.8012 | 3.2412 | 0.5269(0.1021) | 4.2121 | 3.5645 | 0.4899(0.1025) |
| RMR | 0.2201 | 0.2196 | 0.1081(**0.1020**) | 0.2224 | 0.2165 | 0.1108(**0.1029**) | 1.9723 | 1.3241 | 0.3324(0.1022) | 2.1518 | 1.3504 | 0.3451(0.1024) |
| | | | | | | | | | | | | |
| MAM | 0.7981 | 0.7972 | 0.2252(**0.1044**) | 0.7827 | 0.7826 | 0.2155(**0.1039**) | 0.9772 | 0.6943 | 0.2220(0.1038) | 0.9145 | **0.6749** | **0.2218**(**0.1040**) |
| BiGL | 0.8307 | 0.8307 | 0.2318(0.1029) | 0.7998 | 0.7934 | 0.2250(0.1027) | 1.0727 | 0.7525 | 0.2414(0.1030) | 1.0004 | 0.7405 | 0.2365(0.1034) |
| mGAM | 0.7969 | 0.7967 | 0.2249(0.1042) | 0.7787 | 0.7786 | 0.2121(0.1037) | **0.9665** | **0.6842** | **0.2203**(0.1038) | **0.9135** | 0.6753 | 0.2226(0.1037) |
| GroupSpAM | **0.7914** | **0.7916** | **0.2101**(0.1033) | **0.7604** | **0.7599** | **0.2085**(0.1022) | 0.9965 | 0.7142 | 0.2303(**0.1048**) | 0.9521 | 0.7077 | 0.2294(0.1025) |
| GL | 0.8081 | 0.8080 | 0.2241(0.1028) | 0.7741 | 0.7708 | 0.2177(0.1029) | 1.0501 | 0.7306 | 0.2335(0.1030) | 0.9591 | 0.7202 | 0.2363(0.1029) |
| Lasso | 2.5779 | 2.4777 | 0.3956(0.1027) | 2.6801 | 2.6799 | 0.4259(0.1020) | 3.8645 | 3.7612 | 0.5218(0.1020) | 3.7443 | 3.5036 | 0.4963(0.1024) |
| RMR | 1.5160 | 1.5061 | 0.3237(0.1028) | 1.7654 | 1.6798 | 0.3184(0.1020) | 1.9448 | 1.7294 | 0.3587(0.1025) | 2.1085 | 2.0077 | 0.3329(0.1030) |

## 4.2 Coronal mass ejection analysis

Coronal Mass Ejections (CMEs) are the most violent eruptions in the Solar System. It is crucial to forecast the physical parameters related to CMEs. Despite machine learning approaches have been applied to these tasks recently [20, 39], there is no any work for interpretable prediction with data-driven structure discovery. Interplanetary CMEs (ICMEs) data are provided in The Richardson and Cane List (http://www.srl.caltech.edu/ACE/ASC/DATA/level3/icmetable2.htm). From this link, we collect 137 ICMEs observations from 1996 to 2016. The features of CMEs are provided in *SOHO LASCO CME Catalog* (https://cdaw.gsfc.nasa.gov/CME_list/). In-situ solar wind parameters can be downloaded from *OMNIWeb Plus* (https://omniweb.gsfc.nasa.gov/). The in-situ solar wind parameters at earth is used to represent the unknown solar wind plasma [20]. A total of 21 features are chosen as input by combining the features of CMEs and in-situ solar wind parameters. Five physical parameters prediction tasks are considered as outputs including CMEs arrive time, Mean ICME speed, Maximum solar

wind speed, Increment in solar wind speed and Mean magnetic field strength. We split the data of each task into training set, validation set and test set (with ratio $2 : 2 : 1$) and adopt the same settings in simulations. Table 3 demonstrates that MAM enjoy smaller average absolute error than the competitors. In addition, the estimated structure (via MAM) is described in Figure 3. From Figure 3 and Table 3, we know group $G_1$ (including Mass, MPA, Solar wind speed, $V_y$) and group $G_2$ (including Acceleration and Linear Speed) are significant for most tasks. Particularly, $G_2$ and $G_7$ (2nd-order Speed at final height) can be characterized as the factors that reflect the CMEs speed. Table 3 shows that the groups $G_2$ and $G_7$ play an important role in CMEs arrive time prediction, which is consistent with the results in [20]. In addition, the impact of hyper-parameter are displayed in ***Supplementary Material D.2*** due to the space limitation. Overall, the proposed MAM can achieve the promising performance on prediction and structure discovery.

Table 3: Average absolute error and dominant group for each task.

| Tasks | CMEs arrive time | | Mean ICME speed | | Maximum solar wind speed | | Increment in solar wind speed | | Mean magnetic field strength | |
|---|---|---|---|---|---|---|---|---|---|---|
| Methods | AAE $(h)$ | Groups | AAE $(km/s)$ | Groups | AAE $(km/s)$ | Groups | AAE $(km/s)$ | Groups | AAE $(nT)$ | Groups |
| MAM | **9.07** | $G_1,G_2,G_7$ | **45.41** | $G_1,G_2,G_3,G_6$ | 59.32 | $G_1,G_2$ | **65.38** | $G_1,G_2,G_3$ | **3.47** | $G_1$ |
| BiGL | 11.09 | - | 53.75 | - | **46.51** | - | 89.97 | - | 5.21 | - |
| Lasso | 12.16 | - | 62.56 | - | 59.81 | - | 85.34 | - | 4.38 | - |
| RMR | 12.02 | - | 62.23 | - | 51.90 | - | 86.13 | - | 3.98 | - |

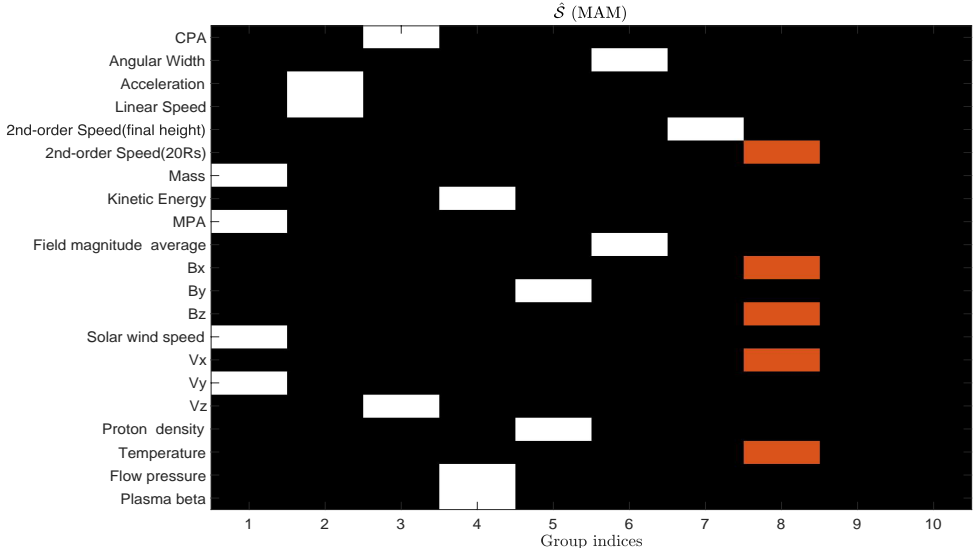

Figure 3: Variable structure $\hat{\mathcal{S}}$ (white pixel=the grouped variables, red pixel=the inactive variables).

## 5   Conclusion

This paper proposes the multi-task additive models to achieve robust estimation and automatic structure discovery. As far as we know, it is novel to explore robust interpretable machine learning by integrating modal regression, additive models and multi-task learning together. The computing algorithm and empirical evaluations are provided to support its effectiveness. In the future, it is interesting to investigate robust additive models for overlapping variable structure discovery [17].

## Broader Impact

The positive impacts of this work are two-fold: 1) Our algorithmic framework paves a new way for mining the intrinsic feature structure among high-dimensional variables, and may be the stepping stone to further explore data-driven structure discovery with overlapping groups. 2) Our MAM can be applied to other fields, e.g, gene expression analysis and drug discovery. However, there is also a risk of resulting an unstable estimation when facing ultra high-dimensional data.

## Acknowledgments

This work was supported by National Natural Science Foundation of China under Grant Nos. 11671161, 12071166, 61972188, 41574181, the Fundamental Research Funds for the Central Universities (Program No. 2662019FW003) and NSERC Grant RGPIN-2016-05024. We are grateful to the anonymous NeurIPS reviewers for their constructive comments.

## Footnotes

*Corresponding author. email: `chenh@mail.hzau.edu.cn`

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
