[Supplementary Material]

# Supplementary Materials to "Multi-task Additive Models for Robust Estimation and Automatic Structure Discovery"

**Yingjie Wang**[1], **Hong Chen**[2,*] **Feng Zheng**[3], **Chen Xu**[4], **Tieliang Gong**[4,5], **Yanhong Chen**[6]
[1]College of Informatics, Huazhong Agricultural University, China
[2] College of Science, Huazhong Agricultural University, China
[3] Department of Computer Science and Engineering,
Southern University of Science and Technology, China
[4] Department of Mathematics and Statistics, University of Ottawa, Canada
[5] School of Computer Science and Technology, Xi'an Jiaotong University, China
[6] National Space Science Center, Chinese Academy of Sciences, China

## A. Notations

Some used notations are summarized in Table 4.

## B. Optimization Details

In this section, we present the details of HQ-DFBB and two partial derivative calculators. Since these calculators are separable w.r.t each task, we only focus on the computation associated with single task by omitting the index $t$, i.e., $h_{\vartheta}(\hat{\beta}(\vartheta), \nu)$ and $h_{\nu}(\hat{\beta}(\vartheta), \nu)$ in the subsequent analysis.

### B.1. HQ-DFBB for inner problem

According to HQ optimization, a convex minimization problem $\min_x Q(x)$ is equivalent to the half-quadratic function $u(x, t)$ with a potential function $s(t)$:

$$\min_{x,t} u(x,t) + s(t),$$

where the dual potential function $s(t)$ can be determined by the convex optimization theory [8]. For a convex function $f(a)$ with its Fenchel conjugate $g(b)$, we have $f(a) = \max_b (ab - g(b))$. Assume that there are a convex function $f(a)$ and a modal kernel $\phi$ satisfying

$$\phi(t/\sigma) = f((t/\sigma)^2) = \max_b ((t/\sigma)^2 b - g(b)), \quad t \in \mathbb{R}. \tag{1}$$

Substituting (1) into the inner problem in Section 2.4, we formulate the transformed inner problem as

$$\min_{\beta,b} \left\{ \frac{1}{n\sigma} \sum_{i=1}^{n} (b_i \big(\frac{y_i - \Psi_i \beta}{\sigma}\big)^2 + g(-b_i)) + \frac{\mu}{2}\|\beta\|_2^2 + \lambda \sum_{l=1}^{L} \tau_l \|\mathcal{T}_{\vartheta_l}\beta\|_2 \right\}, \tag{2}$$

where $b = (b_1, ..., b_n)^T \in \mathbb{R}^n$ is a weight vector with respect to the observations. Problem (2) can be optimized via alternating optimization algorithm.

**Update weight $b$ (given $\beta$)**: According to Theorem 1 in [10], we have

$$b_i = -f'\Big(\Big(\frac{y_i - \Psi_i \beta}{\sigma}\Big)^2\Big), \tag{3}$$

Table 4: Notations

| Notations | Descriptions |
|---|---|
| $\mathcal{X}, \mathcal{Y}$ | the input space and the output space, respectively |
| $X, Y$ | random variables taking values in $\mathcal{X}$ and $\mathcal{Y}$, respectively |
| $x, y$ | realizations of $X$ and $Y$, respectively |
| $P$ | the dimension of input invariables |
| $d$ | the order of spline basis function |
| $n$ | sample size |
| $T$ | the number of tasks |
| $\mathcal{X}^{(t)}, \mathcal{Y}^{(t)}$ | the input space and the output space of $t$-th task, respectively |
| $X^{(t)}, Y^{(t)}$ | random variables taking values in $\mathcal{X}^{(t)}$ and $\mathcal{Y}^{(t)}$, respectively |
| $x^{(t)}, y^{(t)}$ | realizations of $X^{(t)}$ and $Y^{(t)}$, respectively |
| $S^{(t)}$ | the data set of $t$-task with $2n$ sample size, i.e., $S^{(t)} = \{(x_i^{(t)}, y_i^{(t)})\}_{i=1}^{2n}$ |
| $S_{val}^{(t)}$ | a part of $S^{(t)}$ with sample size $n$, which is used for the outer problem |
| $S_{train}^{(t)}$ | a part of $S^{(t)}$ with sample size $n$, which is used for the inner problem |
| $S$ | the union of the data set $S^{(t)}$ of all $T$ tasks, i.e., $S = \{S^{(t)}\}_{t=1}^{T}$ |
| $L$ | the number of group among the variables |
| $G_l$ | the $l$-th ($l \leq L$) group over $\{1, ..., P\}$ |
| $\mathcal{G}^{(t)}$ | the dominant groups of $t$-th task, i.e., $\mathcal{G}^{(t)} \subset \{G_1, ..., G_L\}$ |
| $\mathcal{V}$ | the set of inactive variables |
| $f^*$ | the ground truth function |
| $\phi(\cdot)$ | the representation of modal kernel |
| $\sigma$ | the bandwidth of modal kernel |
| $\mathcal{H}$ | additive hypothesis space $\mathcal{H} = \mathcal{H}_1 \oplus ... \oplus \mathcal{H}_P$ |
| $\bar{\mathcal{B}}$ | the spline-based additive space with infinite order $d$, i.e., $\bar{\mathcal{B}} = \bar{\mathcal{B}}_1 \oplus ... \oplus \bar{\mathcal{B}}_P$ |
| $\mathcal{B}$ | the spline-based additive space with finite order $d$, i.e., $\mathcal{B} = \mathcal{B}_1 \oplus ... \oplus \mathcal{B}_P$ |
| $\mathcal{R}(f)$ | the mode-induced metric for function $f$ |
| $\mathcal{R}^\sigma(f)$ | the KDE-based mode-induced metric for function $f$ |
| $\mathcal{R}_{emp}^\sigma(f)$ | the empirical mode-induced metric for function $f$ |
| $\vartheta \in \mathbb{R}^{P \times L}$ | the representation of group structure |
| $\nu \in \mathbb{R}^P$ | the representation of variable effect |
| $\hat{\mathcal{S}}$ | the inferred variable structure shared by all tasks |
| $\mathcal{S}^*$ | the oracle variable structure shared by all tasks |

where $f'\left(\left(\frac{y_i - \Psi_i \beta}{\sigma}\right)^2\right)$ means the derivative of function $f$ with respect to variable $\left(\frac{y_i - \Psi_i \beta}{\sigma}\right)^2$. For modal kernel functions, we summarize the corresponding function $f(a)$ and weight $b$ in Table 5.

**Update coefficient $\beta$ (given $b$):** Since the proximity operator of $\|\mathcal{T}_{\vartheta_l}\beta\|_2, l \in \{1, ..., L\}$ in (2) cannot be computed in a closed form, the standard forward-backward splitting method [4] cannot be used here directly. In this paper, we apply the forward-backward scheme with Bregman distances (DFBB) to tackle this problem. We first introduce the Fenchel-Rockafellar duality theorem [1]:

**Definition 1.** (Fenchel-Rockafellar duality): *Let $f : \mathcal{X} \to [-\infty, +\infty]$ and $g : \mathcal{Y} \to [-\infty, +\infty]$ be convex functions. Let $\mathcal{T} : \mathcal{X} \to \mathcal{Y}$ be a linear operator with its adjoint operator $\mathcal{T}^*$ such that $< y, \mathcal{T}x > = < \mathcal{T}^* y, x >, \forall x \in \mathcal{X}, \forall y \in \mathcal{Y}$. The primal problem associated with $f$ and $g$ is*

$$\min_x f(x) + g(\mathcal{T}x),$$

*and the dual problem is*

$$\min_y f^*(-\mathcal{T}^* y) + g^*(y).$$

Denote $\odot$ as the Hardamard product and $\Psi = (\Psi_1^T, ..., \Psi_n^T)^T \in \mathbb{R}^{n \times Pd}$. Given weight $b$, let $\widetilde{y} = \sqrt{b} \odot y = (\sqrt{b_1}y_1, ..., \sqrt{b_n}y_n)^T \in \mathbb{R}^n$ and $\widetilde{\Psi} = \sqrt{b} \odot \Psi = (\sqrt{b_1}\Psi_1^T, ..., \sqrt{b_n}\Psi_n^T)^T \in \mathbb{R}^{n \times Pd}$. Set $\varepsilon = n\sigma^3 \mu / 2$ and $\eta = n\sigma^3 \lambda / 2$. Then the transformed inner problem can be rewritten as

$$\min_\beta \left\{ \underbrace{\frac{1}{2}\|\widetilde{y} - \widetilde{\Psi}\beta\|_2^2 + \frac{\varepsilon}{2}\|\beta\|_2^2}_{\mathcal{L}(\beta): \mathbb{R}^{Pd} \to \mathbb{R}} + \underbrace{\eta\|(\tau_l \mathcal{T}_{\vartheta_l}\beta)_{1 \leq l \leq L}\|_2}_{\Omega((\mathcal{T}_{\vartheta_l}\beta)_{1 \leq l \leq L}): \mathbb{R}^{Pd \times L} \to \mathbb{R}} \right\}, \tag{4}$$

Table 5: Definitions and properties of different modal kernel ($t = \frac{y_i - \Psi_i\beta}{\sigma}$, $a \in \mathbb{R}$, $z_1 = e^t + 2 + e^{-t}$, $z_2 = e^t + e^{-t}$ and $z_3 = e^{-t} - e^t$)

|  | Gaussian Kernel | Logistic Kernel | Sigmoid Kernel |
|---|---|---|---|
| $\phi(t)$ | $e^{-\frac{t^2}{2}}$ | $\frac{1}{e^t + 2 + e^{-t}}$ | $\frac{2}{\pi(e^t + e^{-t})}$ |
| $\phi'(t)$ | $-te^{-\frac{t^2}{2}}$ | $\frac{e^{-t} - e^t}{(e^t + 2 + e^{-t})^2}$ | $\frac{2(e^{-t} - e^t)}{\pi(e^t + e^{-t})^2}$ |
| $f(a)$ | $e^{-\frac{a}{2}}$ | $\frac{1}{e^{\sqrt{a}} + 2 + e^{-\sqrt{a}}}$ | $\frac{2}{\pi(e^{\sqrt{a}} + e^{-\sqrt{a}})}$ |
| $b_i$ | $\frac{1}{2}e^{-\frac{t^2}{2}}$ | $\frac{(e^{-t} - e^t)}{2t(e^t + 2 + e^{-t})^2}$ | $\frac{(e^{-t} - e^t)}{\pi t(e^t + e^{-t})^2}$ |
| $(\partial_\beta \mathcal{D}_b)^T$ | $\left(\frac{t}{2}e^{\frac{-t^2}{2}}\Psi_i^T\right)_{1 \leq i \leq N}$ | $\left(\frac{\Psi_i^T[z_2 t z_1^2 + z_1 z_3(2t z_3 + z_1)]}{2\sigma t^2 z_1^4}\right)_{1 \leq i \leq N}$ | $\left(\frac{\Psi_i^T[t z_2^3 + z_2 z_3(2t z_3 + z_1)]}{2\sigma t^2 z_2^4}\right)_{1 \leq i \leq N}$ |

where $\frac{\varepsilon}{2}\|\beta\|_2^2$ assures that $\mathcal{L}(\beta)$ is $\varepsilon$-strongly convex. Denote $w = (w_1, ..., w_L) \in \mathbb{R}^{Pd \times L}$, where $w_l = (w_{(11)l}, ..., w_{(1d)l}, ..., w_{(P1)l}, ..., w_{(Pd)l})^T \in \mathbb{R}^{Pd}$. According to Fenchel-Rockafellar duality theorem, we can formulate the dual problem of of (4) as

$$\hat{w} = \underset{w \in \mathbb{R}^{Pd \times L}}{\arg\min} \left\{ \mathcal{L}^*(-\mathcal{T}_\vartheta^* w) + \Omega^*(w) \right\}. \tag{5}$$

Here $\mathcal{L}^*(-\mathcal{T}_\vartheta^* w)$ is the Fenchel conjugate of $\mathcal{L}(\beta)$ with $\mathcal{T}_\vartheta^* w = \sum_{l=1}^L \mathcal{T}_{\vartheta_l} w_l \in \mathbb{R}^{Pd}$, and $\Omega^*(w) = \sum_{l=1}^L \delta_C(w_l)$ is the Fenchel conjugate of $\Omega(\beta)$, where the indicator function $\delta_C(w_l)$ satisfies $\delta_C(w_l) = 0$ if $\|w_l\|_2 \leq \eta\tau_l$, and $+\infty$ otherwise.

According to the property of strong convex function, $\mathcal{L}^*$ is everywhere differentiable with $\varepsilon^{-1}$-Lipschitz continuous gradient, and $\nabla_w[\mathcal{L}^*(-\mathcal{T}_\vartheta^* w)] = -(\mathcal{T}_{\vartheta_l} \nabla\mathcal{L}^*(-\mathcal{T}_\vartheta^* w))_{1 \leq l \leq L} \in \mathbb{R}^{Pd \times L}$ is $\|(\mathcal{T}_{\vartheta_l}\cdot)_{1 \leq l \leq L}\|_2^2 \varepsilon^{-1}$-Lipschitz continuous.

We next apply the forward-backward splitting with Bregman proximity operator $\text{prox}_{\Omega^*(w)}^\varphi$ to the dual problem (5), where $\varphi$ is the separable Helinger-like function [5, 9], i.e.,

$$\varphi(w) = \sum_{l=1}^L \varphi_l(w_l) = -\sum_{l=1}^L \sqrt{\eta^2\tau_l^2 - \|w_l\|_2^2} \in \mathbb{R}, \ s.t. \ \|w_l\|_2 \leq \eta\tau_l$$

with its Fenchel conjugate

$$\varphi^*(w) = \sum_{l=1}^L \varphi_l^*(w_l) = \sum_{l=1}^L \eta\tau_l\sqrt{1 + \|w_l\|_2^2}.$$

By direct computation, we get

$$\nabla\varphi(w) = \left(\frac{w_l}{\sqrt{\eta^2\tau_l^2 - \|w_l\|_2^2}}\right)_{1 \leq l \leq L}$$

and

$$\nabla\varphi^*(w) = \left(\frac{\eta\tau_l w_l}{\sqrt{1 + \|w_l\|_2^2}}\right)_{1 \leq l \leq L}.$$

Moreover, for any $a = (a_1, ..., a_L) \in \mathbb{R}^{Pd \times L}$,

$$\nabla^2\varphi(w)a = \left(\frac{w_l w_l^T a_l}{(\eta^2\tau_l^2 - \|w_l\|_2^2)^{3/2}} + \frac{a_l}{\sqrt{\eta^2\tau_l^2 - \|w_l\|_2^2}}\right)_{1 \leq l \leq L} \in \mathbb{R}^{Pd \times L}$$

and

$$\nabla^2\varphi^*(w)a = \left(\frac{\eta w_l w_l^T a_l \tau_l}{(1 + \|w_l\|_2^2)^{3/2}} + \frac{\eta\tau_l a_l}{\sqrt{1 + \|w_l\|_2^2}}\right)_{1 \leq l \leq L} \in \mathbb{R}^{Pd \times L}.$$

Denote $q$ as the $q$-th iteration and $\gamma$ as the step-size. To tackle the dual problem (5), we update $w$ by the following iterative step:

$$
\begin{aligned}
w^{(q+1)} &= \mathrm{prox}_{\Omega^*(w)}^{\varphi}\big(\nabla\varphi(w^{(q)}) - \gamma\nabla_w[\mathcal{L}^*(-\mathcal{T}_\vartheta^* w^{(q)})]\big) \\
&= \underset{w \in \mathbb{R}^{Pd \times L}}{\arg\min}\ \varphi(w) + \Omega^*(w) - \big\langle \nabla\varphi(w^{(q)}) - \gamma\nabla_w[\mathcal{L}^*(-\mathcal{T}_\vartheta^* w^{(q)})], w\big\rangle \\
&= \underset{\|w_l\|_2 \leq \eta\tau_l, l=1,\ldots,L}{\arg\min}\ \varphi(w) - \big\langle \nabla\varphi(w^{(q)}) - \gamma\nabla_w[\mathcal{L}^*(-\mathcal{T}_\vartheta^* w^{(q)})], w\big\rangle \\
&= \underset{\|w_l\|_2 \leq \eta\tau_l, l=1,\ldots,L}{\arg\min}\ \sum_{l=1}^{L}\Big\{\varphi_l(w_l) - \big\langle \nabla\varphi_l(w_l^{(q)}) - \gamma\nabla_{w_l}[\mathcal{L}^*(-\mathcal{T}_\vartheta^* w^{(q)})], w_l\big\rangle\Big\} \\
&= \Big(\nabla\varphi_l^*\big(\nabla\varphi_l(w_l^{(q)}) - \gamma\nabla_{w_l}[\mathcal{L}^*(-\mathcal{T}_\vartheta^* w^{(q)})]\big)\Big)_{1 \leq l \leq L} \in \mathbb{R}^{Pd \times L} \\
&= \Big(\nabla\varphi_l^*\big(\nabla\varphi_l(w_l^{(q)}) + \gamma\mathcal{T}_{\vartheta_l}(\widetilde{\Psi}^T\widetilde{\Psi} + \varepsilon\mathbb{I})^{-1}(\widetilde{\Psi}^T\widetilde{y} - \mathcal{T}_\vartheta^* w^{(q)})\big)\Big)_{1 \leq l \leq L}.
\end{aligned}
$$

After $Q$ iterations, we obtain the following primal-dual equation

$$
\beta = \nabla\mathcal{L}^*(-\mathcal{T}_\vartheta^* w^{(Q)}) = (\nabla\mathcal{L}(-\mathcal{T}_\vartheta^* w^{(Q)}))^{-1} = (\widetilde{\Psi}^T\widetilde{\Psi} + \varepsilon\mathbb{I})^{-1}(\widetilde{\Psi}^T\widetilde{y} - \mathcal{T}_\vartheta^* w^{(Q)}).
$$

Now, we summary the HQ-DFBB optimization for the inner problem in Algorithm 2.

---

**Algorithm 2:** HQ-DFBB$(\vartheta, \lambda, \sigma, \mu, \tau; S_{train})$

---

**Input**: Training set $S_{train} = \{(x_i, y_i)\}_{i=1}^{n}$, Hyper- parameter $\vartheta$, Modal kernel $\phi$ with designed function $f$ (Table 5), Bandwidth $\sigma$, Regularization parameter $\lambda > 0$, Penalty parameter $\mu > 0$, weights $\tau_l, l = 1, \ldots, L$.

**Initialization**: Spline basis matrix $\Psi \in \mathbb{R}^{n \times Pd}$ with order $d$, Max-Iter $M, Q \in \mathbb{R}$, $\eta = n\sigma^3\lambda/2$, $\varepsilon = n\sigma^3\mu/2$, $\beta^{(0)} = 0_P$, step-size $\gamma < \varepsilon\eta^{-1}\|(\mathcal{T}_{\vartheta_l}\cdot)_{1 \leq l \leq L}\|_2^{-2}$.

**for** $m = 0 : M - 1$ **do**

   1: Fixed $\beta^{(m)}$, $b^{(m+1)} = \Big(-f'\big(\big(\frac{y_i - \Psi_i\beta^{(m)}}{\sigma}\big)^2\big)\Big)_{1 \leq i \leq n}^{T} \in \mathbb{R}^n$;

   2: Let $\widetilde{y} = \sqrt{b^{(m+1)}} \odot y$, $\widetilde{\Psi} = \sqrt{b^{(m+1)}} \odot \Psi$;

   3: Fixed $b^{(m+1)}$, solve dual problem:

   **Initialization**: $q = 0$, $w^{(m+1,0)} = 0$;

   **for** $q = 0 : Q - 1$ **do**

      $w^{(m+1,q+1)} =$
      $\Big(\nabla\varphi_l^*\big(\nabla\varphi_l(w_l^{(m+1,q)}) + \gamma\mathcal{T}_{\vartheta_l}(\widetilde{\Psi}^T\widetilde{\Psi} + \varepsilon\mathbb{I})^{-1}(\widetilde{\Psi}^T\widetilde{y} - \mathcal{T}_\vartheta^* w^{(m+1,q)})\big)\Big)_{1 \leq l \leq L}$;

   4: $\beta^{(m+1)} = (\widetilde{\Psi}^T\widetilde{\Psi} + \varepsilon\mathbb{I})^{-1}(\widetilde{\Psi}^T\widetilde{y} - \mathcal{T}_\vartheta^* w^{(m+1,Q)})$

**Output:** $\hat{\beta}(\vartheta) = \beta^{(M)}$

---

## B.2. Partial derivative calculator: $h_\vartheta(\hat{\beta}(\vartheta), \nu)$

We calculate the partial derivative calculator $h_\vartheta(\hat{\beta}(\vartheta), \nu)$ based on the backward gradient descent method, *i.e.*,

$$
h_\vartheta(\hat{\beta}(\vartheta), \nu) = -\frac{\partial U(\hat{\beta}(\vartheta), \nu)}{\partial\vartheta} = -\big(\frac{d\hat{\beta}(\vartheta)}{d\vartheta}\big)^T \frac{\partial U(\hat{\beta}(\vartheta), \nu)}{\partial\beta},
$$

where

$$
\frac{\partial U(\hat{\beta}(\vartheta), \nu)}{\partial\beta} = -\frac{1}{n\sigma^2}\sum_{i=1}^{n}\mathcal{T}_\nu\Psi_i^T\phi'\big(\frac{y_i - \Psi_i\mathcal{T}_\nu\hat{\beta}(\vartheta)}{\sigma}\big) \in \mathbb{R}^{Pd}
$$

and $\phi'$ is the derivative of modal kernel (see details in Table 5). We next specify the implementation of the partial derivative. For $m = 0, \ldots, M - 1$ and $q = 0, \ldots, Q - 1$, we denote the equations in Algorithm 2 as

$$
\Big(-f'\big(\big(\frac{y_i - \Psi_i\beta^{(m)}}{\sigma}\big)^2\big)\Big)_{1 \leq i \leq n}^{T} := \mathcal{D}_b(\beta^{(m)}),
$$

$$(\widetilde{\Psi}^T \widetilde{\Psi} + \varepsilon \mathbb{I})^{-1}(\widetilde{\Psi}^T \widetilde{y} - \mathcal{T}_\vartheta^* w^{(m,Q)}) := \mathcal{D}_\beta(b^{(m)}, w^{(m,Q)}, \vartheta),$$

$$\left(\nabla \varphi_l^* \left(\nabla \varphi_l(w_l^{(m,q)}) + \gamma \mathcal{T}_{\vartheta_l}(\widetilde{\Psi}^T \widetilde{\Psi} + \varepsilon \mathbb{I})^{-1}(\widetilde{\Psi}^T \widetilde{y} - \mathcal{T}_\vartheta^* w^{(m+1,q)})\right)\right)_{1 \le l \le L} := \mathcal{D}_w(b^{(m)}, w^{(m,q)}, \vartheta).$$

To illustrate the application of the chain rule, we show a graphical representation of the information

Figure 4: This graph illustrates how the information is back propagated between $\beta^{(m+1)}$ and $\beta^{(m)}$. The derivatives at the nodes show which derivative is to be evaluated from this point downwards through the graph. The edges shows multiplicative factors. The final relationship between $d\beta^{(m+1)}/d\vartheta$ and $d\beta^{(m)}/d\vartheta$ is the sum over all leaf nodes.

flow in Figure 4. For simplicity, we represent the notations in Figure 4 by using the following abbreviations:

$$\mathcal{D}_b(\beta^{(m)}) := \mathcal{D}_b^{(m)}, \quad \mathcal{D}_\beta(b^{(m)}, w^{(m,Q)}, \vartheta) := \mathcal{D}_\beta^{(m,Q)}, \quad \mathcal{D}_w(b^{(m)}, w^{(m,q)}, \vartheta) := \mathcal{D}_w^{(m,q)},$$

with corresponding partial derivatives

$$\frac{d\mathcal{D}_b^{(m)}}{d\beta} := \partial_\beta \mathcal{D}_b^{(m)}, \qquad \frac{\partial \mathcal{D}_\beta^{(m,Q)}}{\partial \vartheta} := \partial_\vartheta \mathcal{D}_\beta^{(m,Q)}, \quad \frac{\partial \mathcal{D}_\beta^{(m,Q)}}{\partial w} := \partial_w \mathcal{D}_\beta^{(m,Q)}, \quad \frac{\partial \mathcal{D}_\beta^{(m,Q)}}{\partial b} := \partial_b \mathcal{D}_\beta^{(m,q)},$$

$$\frac{\partial \mathcal{D}_w^{(m,q)}}{\partial \vartheta} := \partial_\vartheta \mathcal{D}_w^{(m,q)}, \quad \frac{\partial \mathcal{D}_w^{(m,q)}}{\partial w} := \partial_w \mathcal{D}_w^{(m,q)}, \quad \frac{\partial \mathcal{D}_w^{(m,q)}}{\partial b} := \partial_b \mathcal{D}_w^{(m,q)}.$$

From Figure 4, the relationship between $\frac{d\beta^{(m+1)}}{d\vartheta}$ and $\frac{d\beta^{(m)}}{d\vartheta}$ can be represented by

$$\frac{d\beta^{(m+1)}}{d\vartheta} = A_{m+1} + B_{m+1} \frac{d\beta^{(m)}}{d\vartheta},$$

where $A_{m+1}$ is the sum of left leaf nodes and $B_{m+1}$ is the sum of right leaf nodes. Naturally, we have the corresponding transpose

$$(\frac{d\beta^{(m+1)}}{d\vartheta})^T = A_{m+1}^T + (\frac{d\beta^{(m)}}{d\vartheta})^T B_{m+1}^T, \tag{6}$$

where

$$
\begin{aligned}
A_{m+1}^T = {} & (\partial_\vartheta \mathcal{D}_\beta^{(m+1,Q)})^T \\
& + (\partial_\vartheta \mathcal{D}_w^{(m+1,Q-1)})^T (\partial_w \mathcal{D}_\beta^{(m+1,Q)})^T \\
& + (\partial_\vartheta \mathcal{D}_w^{(m+1,Q-2)})^T (\partial_w \mathcal{D}_w^{(m+1,Q-1)})^T (\partial_w \mathcal{D}_\beta^{(m+1,Q)})^T \\
& + \ldots \ldots \\
& + (\partial_\vartheta \mathcal{D}_w^{(m+1,0)})^T (\partial_w \mathcal{D}_w^{(m+1,1)})^T \cdots (\partial_w \mathcal{D}_w^{(m+1,Q-1)})^T (\partial_w \mathcal{D}_\beta^{(m+1,Q)})^T
\end{aligned}
\tag{7}
$$

and

$$
\begin{aligned}
B_{m+1}^T = {} & (\partial_\beta \mathcal{D}_b^{(m)})^T (\partial_b \mathcal{D}_\beta^{(m+1,Q)})^T \\
& + (\partial_\beta \mathcal{D}_b^{(m)})^T (\partial_b \mathcal{D}_w^{(m+1,Q-1)})^T (\partial_w \mathcal{D}_\beta^{(m+1,Q)})^T \\
& + \ldots \ldots \\
& + (\partial_\beta \mathcal{D}_b^{(m)})^T (\partial_b \mathcal{D}_w^{(m+1,0)})^T (\partial_w \mathcal{D}_w^{(m+1,1)})^T \cdots (\partial_w \mathcal{D}_\beta^{(m+1,Q)})^T.
\end{aligned}
\tag{8}
$$

We now specialize the computation of $A_m^T$ and $B_m^T$, for every $m = 1, ..., M$. Denote $\Psi = (\bar{\Psi}_1, ..., \bar{\Psi}_{Pd}) \in \mathbb{R}^{n \times Pd}$, where each $\bar{\Psi}_j = (\psi_j(x_{1j}), ..., \psi_j(x_{nj}))^T \in \mathbb{R}^n$, $j = 1, ..., P$. Denote $\mathcal{T}_{\bar{\Psi}_j} b = \bar{\Psi}_j \odot b \in \mathbb{R}^n$ and $u = (\widetilde{\Psi}^T \widetilde{\Psi} + \varepsilon \mathbb{I}) \in \mathbb{R}^{Pd \times Pd}$. Then we get

$$
(\partial_b \mathcal{D}_\beta^{(m,Q)})^T = u^{-1} (\mathcal{T}_{\bar{\Psi}_j} b^{(m)})_{1 \le j \le Pd}^T + \left( u^{-1} \widetilde{\Psi}_i^T \widetilde{\Psi}_i u^{-1} (\widetilde{\Psi}^T \widetilde{y} - \mathcal{T}_\vartheta^* w^{(m,Q)}) \right)_{1 \le i \le n}.
\tag{9}
$$

For each $a = (a_1, ..., a_L) \in \mathbb{R}^{Pd \times L}$, there hold

$$
\begin{aligned}
(\partial_w \mathcal{D}_\beta^{(m,Q)})^T a &= \left( \mathcal{T}_{\vartheta_l} (u^{-1} a_l) \right)_{1 \le l \le L} \in \mathbb{R}^{Pd \times L} \\
(\partial_\vartheta \mathcal{D}_\beta^{(m,Q)})^T a &= \left( \mathcal{T}_{w_l^{(m,Q)}} (u^{-1} a_l) \right)_{1 \le l \le L} \in \mathbb{R}^{P \times L}.
\end{aligned}
\tag{10}
$$

Note that $\mathcal{T}_{w_l} a = (\sum_{t=1}^d w_{(jt)l} a_{jt})_{1 \le j \le P}^T$ for every $a = (a_{11}, .., a_{1d}, ..., a_{P1}, ..., a_{Pd})^T \in \mathbb{R}^{Pd}$, $j \in \{1, ..., P\}$. Let $v_l = \nabla \varphi_l(w_l^{(m,q)}) + \gamma \mathcal{T}_{\vartheta_l} \mathcal{D}_\beta^{(m,Q)}$. For every $a = (a_1, ..., a_L) \in \mathbb{R}^{Pd \times L}$, we have

$$
\begin{aligned}
(\partial_b \mathcal{D}_w^{(m,q)})^T a &= \gamma (\partial_b \mathcal{D}_\beta^{(m,q)})^T \mathcal{T}_\vartheta^* \nabla^2 \varphi^*(v) a, \\
(\partial_w \mathcal{D}_w^{(m,q)})^T a &= \left( \nabla^2 \varphi(w_l^{(m,q)}) \nabla^2 \varphi^*(v_l) a_l - \gamma (\partial_w \mathcal{D}_\beta^{(m,q)})^T \mathcal{T}_\vartheta^* \nabla^2 \varphi^*(v) a \right)_{1 \le l \le L}, \\
(\partial_\vartheta \mathcal{D}_w^{(m,q)})^T a &= \gamma \left( (\partial_\vartheta \mathcal{D}_\beta^{(m,Q)})^T \mathcal{T}_\vartheta^* \nabla \varphi^*(v) a + \mathcal{T}_{\nabla \varphi_l^*(v_l) a_l} \mathcal{D}_\beta^{(m,Q)} \right)_{1 \le l \le L}.
\end{aligned}
$$

In addition, the partial derivative $(\partial_\beta \mathcal{D}_b^{(m)})^T$ in (8) is summarized in Table 5. Combining the above computations, we have

$$
\begin{aligned}
\frac{\partial U(\hat{\beta}(\vartheta), \nu)}{\partial \vartheta} &= (\frac{d\hat{\beta}^{(M)}(\vartheta)}{d\vartheta})^T \underbrace{\frac{\partial U(\hat{\beta}(\vartheta), \nu)}{\partial \beta}}_{D_M} \\
&= \underbrace{A_M^T D_M}_{C_{M-1}} + (\frac{d\hat{\beta}^{(M-1)}(\vartheta)}{d\vartheta})^T \underbrace{B_M^T D_M}_{D_{M-1}} \\
&= \underbrace{C_{M-1} + A_{M-1}^T D_{M-1}}_{C_{M-2}} + (\frac{d\hat{\beta}^{(M-2)}(\vartheta)}{d\vartheta})^T \underbrace{B_{M-1}^T D_{M-1}}_{D_{M-2}} \\
&= \ldots \\
&= \underbrace{C_{M-1} + A_{M-1}^T D_{M-1} + A_{M-2}^T D_{M-2} + \ldots + A_1^T D_1}_{C_0}.
\end{aligned}
\tag{11}
$$

Now, we state the computing steps for the partial derivative with respect to variable $\vartheta$ as below.

---

**Algorithm 3:** Partial Derivative Calculator $h_\vartheta(\hat{\beta}(\vartheta), \nu)$

---

**Input**: Training set $S_{train} = \{(x_i, y_i)\}_{i=1}^n$, Validation set $S_{val} = \{(x_i, y_i)\}_{i=1}^n$,
  Hyper-parameter $\vartheta$, Max-Iter $M, Q \in \mathbb{R}$, Modal kernel $\phi$, Bandwidth $\sigma$, Regularization
  parameter $\lambda$, Penalty parameter $\mu$.
**Initialization**: Spline basis $\Psi \in \mathbb{R}^{n \times Pd}$ with order $d$ for validation set, $\eta = m\sigma^3\lambda/2$,
  $\varepsilon = n\sigma^3\mu$, $\hat{\beta}^{(0)} = 0_P$, step-size $\gamma < \epsilon\lambda^{-1}\|(\mathcal{T}_{\vartheta_l}\cdot)_{l=1}^L\|_2^2$.
**1. Solve the inner problem by HQ-DFBB based on training set $S_{train}$ (Algorithm 2):**
     Output: $\beta^{(m)}, b^{(m)}, w^{(m,q)}, \forall m \in \{1, ..., M\}, \forall q \in \{1, ..., Q\}$
**2. Compute partial derivative based on validation set $S_{val}$ with spline basis $\Psi$:**

**Initialization:** $D_M = \frac{\partial U(\beta^{(M)}(\vartheta), \nu)}{\partial \beta}, C_M = 0$

**for** $m = M : -1 : 1$ **do**
  $\quad$ 1): Compute $A_m^T$ by Equation (7)
  $\quad$ 2): Compute $B_m^T$ by Equation (8)
  $\quad$ 3): $C_{m-1} = A_m^T D_m + C_m$
  $\quad$ 4): $D_{m-1} = B_m^T D_m$
**Output:** $C_0$

---

### B.3. Partial derivative calculator $h_\nu(\hat{\beta}(\vartheta), \nu)$

Since variable $\nu$ appears explicitly in the outer problem, we can obtain $\partial U(\hat{\beta}(\vartheta), \nu)/\partial\nu, t = 1, ..., T$ directly. Given the solution $\hat{\beta}(\vartheta)$ (also $\beta^{(M)}$) of the inner problem, we have

$$h_\nu(\hat{\beta}(\vartheta), \nu) = -\frac{\partial U(\hat{\beta}(\vartheta), \nu)}{\partial\nu} = \frac{1}{n\sigma^2}\sum_{i=1}^n \phi'(\frac{y_i - \Psi_i \mathcal{T}_\nu \hat{\beta}(\vartheta)}{\sigma})\mathcal{T}_{\Psi_i^T}\hat{\beta}(\vartheta),$$

where

$$\mathcal{T}_{\Psi_i^T}\hat{\beta}(\vartheta) = \big(\sum_{t=1}^d \psi_{1t}(x_{i1})^T\hat{\beta}_{1t}(\vartheta), ..., \sum_{t=1}^d \psi_{Pt}(x_{iP})^T\hat{\beta}_{Pt}(\vartheta)\big)^T \in \mathbb{R}^P.$$

## C. Convergence Analysis of Optimization Algorithm

From Theorem 2.1 in [5] and Theorem 4 in [7], we know that the Algorithm 1 in Section 3 converges only if the iteration sequence generated by HQ-DFBB in Algorithm 2 converges to the solution of the inner problem. Since the analysis result is suitable to every task, we only focus on a single task by omitting the index $t$.

Denote $J(\beta)$ as the objective function of inner problem and $J(\beta, b)$ as the transformed inner objective function (4) by omitting index $t$. From HQ optimization, we know that

$$J(\beta) = 2n^{-1}\sigma^{-3}\min_b J(\beta, b), \ \ \forall\beta \in \mathbb{R}^{Pd}. \tag{12}$$

Due to $\varepsilon$-strongly convex, $J(\beta, b)$ has a unique global minimum for a given $b \in \mathbb{R}^n$.

HQ-DFBB is formulated with $Q$ inner loops and $M$ outer loops. For every $m \in \{0, ..., M-1\}$, we denote $\bar{\beta}^{(m+1)} = \arg\min_\beta J(\beta, \bar{b}^{(m+1)})$ and $\bar{b}^{(m+1)} = \arg\min_b J(\bar{\beta}^{(m)}, b)$. For given outer iteration times $M \in \mathbb{N}$, $\beta^{(M)}$ is the solution of HQ-DFBB and is dependent on the inner iteration $Q$. Let $\lim_{Q,M\to+\infty} \beta^{(M)} = \beta^*$, $\lim_{Q,M\to+\infty} b^{(M)} = b^*$.

To get the convergence analysis for the computing algorithm, we introduce the following result established in [5].

**Lemma 1.** *For any step-size $\gamma < \varepsilon\lambda^{-1}\|(\mathcal{T}_{\vartheta_l}\cdot)_{1\le l\le L}\|_2^2$, the sequence $\{\beta^{(m+1)}\}_{Q\in\mathbb{N}}$ converges to $\bar{\beta}^{(m+1)}$. If $\gamma = \frac{1}{2}\varepsilon\lambda^{-1}\|(\mathcal{T}_{\vartheta_l}\cdot)_{1\le l\le L}\|_2^2$, there holds*

$$\frac{1}{2}\|\beta^{(m+1)} - \bar{\beta}^{(m+1)}\|_2^2 \le \frac{2\eta}{Q\varepsilon^2}\|\mathcal{T}_{\vartheta_l}\|^2 D_\Phi(\omega, \omega^{(0)}), \tag{13}$$

where $D_\Phi$ is the Bregman distance associated to $\Phi$, $\|\mathcal{T}_{\vartheta_l}\|^2 D_\Phi(\omega, \omega^{(0)})$ is uniformly bounded from above on $\Theta$. As a result, there exists a constant $C > 0$, such that

$$\|\beta^{(m+1)} - \bar{\beta}^{(m+1)}\|_2^2 \leq \frac{C}{Q}. \tag{14}$$

It is a position to state the convergence guarantees for our HQ-DFBB in Algorithm 2.

**Theorem 1.** *If weight $b$ in (3) is a continuous differential function with respect to $\beta$, as $Q, M \to +\infty$, there hold*

*(a) The function sequences $\{J(\beta^{(m)}, b^{(m)}), m = 1, ..., M\}$ and $\{J(\beta^{(m)}), m = 1, ..., M\}$ converge,*

*(b) $\|\beta^{(M)} - \beta^{(M-1)}\|_2^2 \to 0$,*

*(c) $\beta^* = \arg\min_\beta J(\beta, b^*)$.*

*Proof.* (a) Since $J(\beta, b)$ is strong convex w.r.t. $\beta$, then

$$J(\bar{\beta}^{(m+1)}, \bar{b}^{(m+1)}) < J(\bar{\beta}^{(m)}, \bar{b}^{(m+1)}) \leq J(\bar{\beta}^{(m)}, \bar{b}^{(m)}). \tag{15}$$

According to Lemma 1, we have

$$\lim_{Q \to +\infty} \beta^{(m+1)} = \bar{\beta}^{(m+1)}.$$

Let $b^{(m+1)} = \arg\min_b J(\beta^{(Q,m)}, b)$. Due to $J(\beta, b)$ and $b$ are continuous functions with respect to $\beta$, for any $m \in \{0, ..., M-1\}$, we have

$$\bar{b}^{(m+1)} = \arg\min_b \lim_{Q \to +\infty} J(\beta^{(m)}, b).$$

Then,

$$\lim_{Q \to +\infty} J(\beta^{(m+1)}, b^{(m+1)}) = J(\bar{\beta}^{(m+1)}, \bar{b}^{(m+1)}), \quad \lim_{Q \to +\infty} J(\beta^{(m)}, b^{(m+1)}) = J(\bar{\beta}^{(m)}, \bar{b}^{(m+1)}). \tag{16}$$

According to the order-preserving property of limit, there exists a large enough $Q \in \mathbb{N}$ such that

$$J(\beta^{(m+1)}, b^{(m+1)}) < J(\beta^{(m)}, b^{(m+1)}). \tag{17}$$

Moreover, there exists a large enough $Q \in \mathbb{N}$ such that

$$J(\beta^{(m)}, b^{(m+1)}) \leq J(\beta^{(m)}, b^{(m)}).$$

As a result, $\{J(\beta^{(m)}, b^{(m)}), m = 1, 2, ...\}$ is a decreasing sequence. Since $J(\beta, b)$ is bounded below, $\{J(\beta^{(m)}, b^{(m)}), m = 1, 2, ...\}$ converges. According to $J(\beta) = \min_b J(\beta, b)$ in (12), we obtain $J(\beta^{(m)}) = n^{-1}\sigma^{-3} J(\beta^{(m)}, b^{(m+1)})$. Naturally, the sequence $\{J(\beta^{(m)}), m = 1, 2, ...\}$ also converges.

(b) Since $J(\beta, b)$ is a $\varepsilon$-strongly convex function w.r.t $\beta$, we get

$$J(\bar{\beta}^{(m+1)}, \bar{b}^{(m)}) - J(\bar{\beta}^{(m)}, \bar{b}^{(m)}) \geq g^T(\bar{\beta}^{(m+1)} - \bar{\beta}^{(m)}) + \frac{\varepsilon}{2}\|\bar{\beta}^{(m+1)} - \bar{\beta}^{(m)}\|^2,$$

where $g$ denotes any gradient of $J(\bar{\beta}^{(m)}, \bar{b}^{(m)})$. Due to $\bar{\beta}^{(m)} = \arg\min_\beta J(\beta, \bar{b}^{(m)})$, we can take $g = 0$. Then,

$$\frac{\varepsilon}{2}\|\bar{\beta}^{(m+1)} - \bar{\beta}^{(m)}\|^2 \leq J(\bar{\beta}^{(m+1)}, \bar{b}^{(m)}) - J(\bar{\beta}^{(m)}, \bar{b}^{(m)}). \tag{18}$$

Considering $J(\bar{\beta}^{(m+1)}, \bar{b}^{(m)}) - J(\bar{\beta}^{(m)}, \bar{b}^{(m)}) \to 0$ as $m \to +\infty$, we can deduce from (18) that $\|\bar{\beta}^{(m+1)} - \bar{\beta}^{(m)}\|^2 \to 0$ as $m \to +\infty$. Moreover, from Lemma 1, we have

$$\|\beta^{(Q,m+1)} - \beta^{(Q,m)}\|^2 \leq \|\beta^{(Q,m+1)} - \bar{\beta}^{(m+1)}\|^2 + \|\bar{\beta}^{(m+1)} - \bar{\beta}^{(m)}\|^2 + \|\bar{\beta}^{(m)} - \beta^{(Q,m)}\|^2$$

$$\leq \frac{2C}{Q} + \|\bar{\beta}^{(m+1)} - \bar{\beta}^{(m)}\|^2.$$

The desired result in (b) follows by taking $m, Q \to +\infty$.

(c) Since the weight $b$ is continuous differential with respect to $\beta$, we have $b^{(m)}$ is convergent if $\beta^{(m)}$ is convergent as $Q, m \to +\infty$. Based on Theorem 3.1 in [9], the sequence $\{w^{(m,Q)}\}_{Q \in \mathbb{N}}$ generated in Algorithm 2 is convergent to $\hat{w}^{(m)}$ as $Q \to +\infty$. Let $\lim_{m \to +\infty} \hat{w}^{(m)} = \hat{w}^*$, $\lim_{Q,m \to +\infty} \beta^{(m)} = \beta^*$, $\lim_{Q,m \to +\infty} b^{(m)} = b^*$, $\widetilde{\Psi}^* = \sqrt{b^*} \odot \Psi$ and $\widetilde{y}^* = \sqrt{b^*} \odot y$. According to the following primal-dual link

$$\beta^{(m+1)} = (\widetilde{\Psi}^T \widetilde{\Psi} + \varepsilon \mathbb{I})^{-1}(\widetilde{\Psi}^T \widetilde{y} - \mathcal{T}_\vartheta^* w^{(m+1,Q)}),$$

we have

$$\beta^* = (\widetilde{\Psi}^{*T} \widetilde{\Psi}^* + \varepsilon \mathbb{I})^{-1}(\widetilde{\Psi}^{*T} \widetilde{y}^* - \mathcal{T}_\vartheta^* \hat{w}^*), \quad \text{as } Q, m \to +\infty. \tag{19}$$

By direct computation, (19) can be rewritten as

$$(\widetilde{\Psi}^{*T} \widetilde{\Psi}^* + \varepsilon \mathbb{I})\beta^* - \widetilde{\Psi}^{*T} \widetilde{y}^* + \mathcal{T}_\vartheta^* \hat{w}^* = 0.$$

From the foward-backward iteration process, we know $\|w_l^{(m,q)}\| \leq \eta \tau_l$ for every $m \in \{1, ..., M\}, l \in \{1, ..., L\}, q \in \{1, ..., Q\}$. As a result, we get $\|\hat{w}_l^*\| \leq \eta \tau_l$ by taking $m, Q \to +\infty$. Denote $w_l^* = \frac{1}{\eta \tau_l} \hat{w}_l^*$ such that $\|w_l^*\| \leq 1$. It follows that

$$(\widetilde{\Psi}^{*T} \widetilde{\Psi}^* + \varepsilon \mathbb{I})\beta^* - \widetilde{\Psi}^{*T} \widetilde{y}^* + \mathcal{T}_\vartheta^* \hat{w} = (\widetilde{\Psi}^{*T} \widetilde{\Psi}^* + \varepsilon \mathbb{I})\beta^* - \widetilde{\Psi}^{*T} \widetilde{y}^* + \sum_{l=1}^{L} \eta \tau_l \mathcal{T}_{\vartheta_l} w_l^* = 0. \tag{20}$$

From the definition of $\Omega(\beta)$ in (4), we derive

$$\partial \Omega(\beta) = \eta \sum_{l=1}^{L} \tau_l \mathcal{T}_{\vartheta_l} \delta_l,$$

where $\delta_l, 1 \leq l \leq L$ satisfies $\|\delta_l\| = 1$ if $\mathcal{T}_{\vartheta_l} \beta \neq 0$, and $\|\delta_l\| < 1$ otherwise. Therefore

$$0 = (\widetilde{\Psi}^{*T} \widetilde{\Psi}^* + \varepsilon \mathbb{I})\beta^* - \widetilde{\Psi}^{*T} \widetilde{y}^* + \sum_{l=1}^{L} \eta \tau_l \mathcal{T}_{\vartheta_l} w_l^* \in \nabla \mathcal{L}(\beta^*) + \partial \Omega(\beta^*).$$

That is to say $0 \in \partial_\beta J(\beta^*, b^*)$. Since $J(\beta, b)$ is strongly convex with respect to $\beta$, we know $\beta^*$ is the unique solution of $J(\beta, b^*)$. $\qquad\square$

**Remark 1.** *Theorem 1 illustrates the convergence of HQ-DFBB in Algorithm 2. Combining Theorem 1 with Theorem 2.1 and Theorem 3.2 in [5], we obtain the convergence of Algorithm 1 in Section 3. However, to solve the non-convex inner problem, our optimization strategy have extral computation time and space complexity compared with [5]. For instance, the half-quadratic optimization is introduced to solve the nonconvex inner problem, which results in more computation time for our optimization strategy because of the additional outer loop in Algorithm 2.*

## D. Experiments

### D.1. Simulated data analysis

In this section, we firstly give the remaining results associated with Chi-square noise and Student noise in Table 6 and Figures 5-7. To investigate the impact of hyper-parameters on MAM, some additional evaluations, exemplified by Example A (Gaussian noise and $|\mathcal{V}| = 0$), are provided as below:

**Impact of the number of groups**: In previous evaluations, the number of groups was known a priori, i.e., $L = 5$. We relax this assumption and implement our method with a larger number of groups, i.e., $L = 10$. Figures 8 (top right panel) shows that the 5 extra groups are empty. It indicates that a satisfactory inference also can be obtained when the number of groups is set to be larger than the oracle number of groups.

**Impact of outer iterations $Z$ and parameter $\lambda$**: For various values of regularization parameter $\lambda = 10^{-4}, 10^{-3}, 10^{-2}, 10^{-1}$, Figures 8 (bottom left panel) shows the outer objective increases until converges as the number of outer iterations $Z$ grows.

**Impact of inner iterations $Q$ and $M$**: We further investigate the impact of the number of inner iteration $Q$ and $M$ on the validation error. To do so, we repeat the same experiment for different values of $Q = 100, 200, 400, 800$ and $M = 1, 2, 4, 8, 16$. The results in Figures 8 (top right panel)

indicate that increasing $Q$ and $M$ sufficiently permits yielding smaller TD.

**Impact of the bandwidth $\sigma$**: To investigate the sensitivity of MAM towards the choice of bandwidths, we repeat the same experiment for different bandwidths $\sigma = 1, 2, 4, 8, 16$. Figures 8 (bottom right panel) shows that the performance of MAM gets close to mGAM as $\sigma$ decreases.

Table 6: Performance comparisons on Example A (top) and Example B (bottom) w.r.t different criterions.

| Methods | $|\mathcal{V}| = 0$ (Chi-square noise) | | | $|\mathcal{V}| = 5$ (Chi-square noise) | | | $|\mathcal{V}| = 0$ (Exponential noise) | | | $|\mathcal{V}| = 5$ (Exponential noise) | | |
|---|---|---|---|---|---|---|---|---|---|---|---|---|
| | ASE | TD | WPI (SCP) | ASE | TD | WPI (SCP) | ASE | TD | WPI (SCP) | ASE | TD | WPI (SCP) |
| MAM | 0.8114 | 0.7982 | 0.2075(**0.1019**) | 0.8212 | 0.8184 | 0.2079(0.1015) | **0.8124** | **0.7894** | 0.2092(**0.1025**) | 0.8373 | 0.8129 | **0.2080**(0.1016) |
| BiGL | 0.8908 | 0.8848 | 0.2182(0.1015) | 0.8908 | 0.8594 | 0.2217(0.1017) | 0.8958 | 0.8643 | 0.2268(0.1022) | 0.8813 | 0.8648 | 0.2289(0.1018) |
| mGAM | **0.8091** | **0.7887** | **0.2034**(0.1016) | **0.8118** | **0.8060** | **0.2052**(0.1016) | 0.8147 | 0.7902 | **0.2011**(0.1023) | **0.8316** | **0.8083** | 0.2143(0.1018) |
| GL | 0.8642 | 0.8581 | 0.2139(0.1017) | 0.8754 | 0.8391 | 0.2107(**0.1022**) | 0.8729 | 0.8413 | 0.2231(0.1014) | 0.8625 | 0.8420 | 0.2220(0.1020) |
| Lasso | 3.6536 | 3.6364 | 0.4982(0.1018) | 3.5545 | 3.5364 | 0.4926(0.1020) | 3.5823 | 3.5798 | 0.5012(0.1020) | 3.8042 | 3.8033 | 0.5192(0.1022) |
| RMR | 1.8196 | 1.8158 | 0.3614(0.1016) | 1.9145 | 1.8968 | 0.3733(0.1017) | 2.0576 | 1.8686 | 0.3601(0.1024) | 1.9875 | 1.9377 | 0.3574(**0.1023**) |
| MAM | 0.8424 | 0.8356 | **0.2264**(**0.1038**) | 0.8229 | 0.8196 | **0.2383**(0.1033) | 0.8414 | 0.8294 | **0.2314**(**0.1045**) | 0.8331 | 0.8209 | 0.2292(0.1043) |
| BiGL | 0.8982 | 0.8813 | 0.2327(0.1034) | 0.8994 | 0.8994 | 0.2489(0.1026) | 0.9007 | 0.9006 | 0.2420(0.1036) | 0.8919 | 0.8975 | 0.2509(0.1036) |
| mGAM | **0.8411** | **0.8322** | 0.2291(0.1034) | **0.8196** | **0.8013** | 0.2406(**0.1036**) | **0.8318** | **0.8163** | 0.2323(0.1035) | **0.8308** | **0.8161** | **0.2261**(**0.1044**) |
| GroupSpAM | 0.8677 | 0.8555 | 0.2299(0.1029) | 0.8489 | 0.8302 | 0.2477(0.1033) | 0.8742 | 0.8602 | 0.2433(0.1029) | 0.8599 | 0.8574 | 0.2277(0.1033) |
| GL | 0.8843 | 0.8804 | 0.2279(0.1032) | 0.8786 | 0.8778 | 0.2480(0.1028) | 0.8856 | 0.8857 | 0.2441(0.1028) | 0.8621 | 0.8672 | 0.2431(0.1031) |
| Lasso | 3.4796 | 3.4313 | 0.4761(0.1025) | 3.3763 | 3.3251 | 0.4679(0.1025) | 3.3722 | 3.3618 | 0.4881(0.1023) | 3.3537 | 3.3216 | 0.4778(0.1033) |
| RMR | 1.9286 | 1.7885 | 0.3594(0.1031) | 1.7860 | 1.7843 | 0.3506(0.1028) | 1.8451 | 1.8241 | 0.3505(0.1026) | 1.8920 | 1.8971 | 0.3622(0.1021) |

Figure 5: Variable structure discovery for *Example B* under different noise and sparsity index $\mathcal{V}$ (white pixel means the grouped variables and red pixel means the inactive variables). Top left panel: *Gaussian noise* and $|\mathcal{V}| = 0$. Top right panel: *Student noise* and $|\mathcal{V}| = 0$. Bottom left panel: *Gaussian noise* and $|\mathcal{V}| = 5$. Bottom right panel: *Student noise* and $|\mathcal{V}| = 5$.

### D.2. Coronal mass ejection analysis

CMEs data contain 137 observations with 21 variables including: (1) Central PA (CPA), (2) Angular Width, (3)Acceleration, three approximated speeds ((4) Linear Speed, (5) 2nd-order Speed at final height, (6) 2nd-order Speed at 20 Rs), (7) Mass, (8) Kinetic Energy, (9) MPA, (10) Field magnitude average, (11) $B_x$, (12) $B_y$, (13) $B_z$, (14) Solar wind speed, (15) $V_x$, (16) $V_y$, (17) $V_z$, (18) Proton density, (19) Temperature, (20) Flow pressure and (21) Plasma beta. Correspondingly, the outputs (tasks) include CMEs arrive time, Mean ICME speed, Maximum solar wind speed, Increment in solar wind speed and Mean magnetic field strength.

Figures 9 shows the impact of the hyper-parameters (the number of groups and regularization parameter $\lambda$) on the average absolute error (AAE) for each task.

Figure 6: Variable structure discovery for *Example A* under different noise and sparsity index $\mathcal{V}$ (white pixel means the grouped variables and red pixel means the inactive variables). Top left panel: *Chi-square noise* and $|\mathcal{V}| = 0$. Top right panel: *Exponential noise* and $|\mathcal{V}| = 0$. Bottom left panel: *Chi-square noise* and $|\mathcal{V}| = 5$. Bottom right panel: *Exponential noise* and $|\mathcal{V}| = 5$.

Figure 7: Variable structure discovery for *Example B* under different noise and sparsity index $\mathcal{V}$ (white pixel means the grouped variables and red pixel means the inactive variables). Top left panel: *Chi-square noise* and $|\mathcal{V}| = 0$. Top right panel: *Exponential noise* and $|\mathcal{V}| = 0$. Bottom left panel: *Chi-square noise* and $|\mathcal{V}| = 5$. Bottom right panel: *Exponential noise* and $|\mathcal{V}| = 5$.

# E. Discussion

This paper proposes the MAM and evaluate its effectiveness on regression prediction and structure discovery. Now, we outline some future research directions related closely with our MAM.

## E.1 Generalization error analysis based on algorithmic stability

In theory, it is valuable to provide some generalization ability analysis of MAM. The key challenges for theoretical analysis are the non-convex mode-induced metric and nonlinear additive hypothesis space. With the help of algorithmic stability in [2, 11], we can establish the generalization error bound of multi-task additive models with some additional restrictions on error metric, e.g., Quadratic Growth condition [3]. In the future, it is important to relax the restrictions on error metric.

Figure 8: The impact of hyper-parameters on MAM. Top left panel: a satisfactory estimation of the groups can be obtained even when the number of groups is set to $L = 10$ instead of the oracle number of groups $L = 5$. Bottom left panel: the maximization of $\mathcal{U}(\beta(\vartheta^{(z)}), \nu^{(z)})$ is displayed for various regularization parameters $\lambda$. Top right panel: the TD is displayed for various $M$ and $Q$. Bottom right panel: the TD is displayed for various bandwidth $\sigma$.

Figure 9: The impact of the regularization parameter $\lambda$ and the number of groups on $\mathcal{U}(\hat{\beta}(\hat{\vartheta}), \hat{\nu})$ and average absolute error (AAE).

### E.2 Overlapping variable structure discovery

It should be noticed that our MAM is limited to the automatic structure discovery without overlapping groups. That is to say, each variable in the estimated structure $\hat{S}$ can only be assigned into one group according to the property of unit simplex $\sum_{l=1}^{L} \vartheta_{jl} = 1$. Some attempts have been made to achieve promising estimation performance with a given overlapping group structure [6]. It is interesting to extend our MAM to mine the overlapping structure automatically without priori knowledge.

## Footnotes

*Corresponding author. email: `chenh@mail.hzau.edu.cn`