[Reviews · NeurIPS 2020]

Review 1

Summary and Contributions: The paper introduces a multi-task additive model for automatic discovery of variable structure when the involved data has complex non-Gaussian noise. The method is formalized as a bi-level optimization problem where the outer problem is an empirical version of the mode-induced metric using KDE and the inner problem is the regularized empirical risk minimization problem that finds the group variable structure. The authors solve the optimization problem using Prox-SAGA with projection into the compact domains (simplex and infinity ball), where they propagate the gradients through the argmin using a smooth iterative algorithm for the inner problem. They provide a comparative experimental analysis with existing baselines in a synthetic dataset and an evaluation on real data of Coronal Mass Ejection (CME).

Strengths: - The method combines successfully a multi-task additive model, robust estimation and automatic structure discovery into a single framework. - They provide a convergent algorithm to compute the structure. - The comparative analysis of section 4.1 shows that the method outperforms the baselines for non Gaussian noise.

Weaknesses: - METHOD: My main concern about this paper is on the scientific novelty of this work and the motivation to combine both robust estimation and automatic structure discovery. If we remove the mode-induced metric and replace it with the least-square loss, the method is (essentially) the same as the one of [11] (the only difference would be the mask to screen main effect variables, idea extracted from [34]). Adding the mode-induced metric can be easily combined to the framework by using results from [37]. In other words, I feel that this paper is just a combination of the bi-level module of [11] for automatic structure discovery and the regularized mode-induced metric estimator of [37], which can be both easily combined by reading through these papers. This could be justified if there was a strong underlying motivation to justify the combination of these ideas, but I don't think this is the case. Can the authors comment on the deep interest of combining both automatic structure discovery and robust estimation beyond the fact that they can just be readily combined from previous work? - ALGORITHM: If I am not wrong, the empirical mode-induced metric R_{emp}^\sigma is *non-convex*, which is not discussed at all in the paper and it is a major challenge compared to the conditional mean estimator which is computed from the least-square loss. Is this correct? I carefully checked the convergence analysis and Remark 3 seems correct: indeed, there is no statement whatsoever about reaching a global minimum but just a statement of convergence and Theorem 2.1 of [11] does not need the convexity of the outter objective, only smoothness. Nevertheless, it is very important to highlight the extra layer of complexity (maybe hardness to reach a better local optimum) that the non-convexity of the mode-induced metric brings compared to least-square loss. Should one be more pessimistic about the quality of the solution because of the non-convexity of the mode-induced metric (compared to the least-square loss)?

Correctness: - EVALUATION: There is no comparison to any baseline for the experiment of Coronal Mass Ejection in section 4.2., so the results are very difficult to assess. Why haven't you compared the method to other baselines for variable structure discovery? If there is no comparison, the average absolute errors in Table 3 are non-informative of the performance of the method and there is no way to make such evaluation.

Clarity: The paper is in general well-written. Some comments: - It is hard to read through subsections 2.3 and 2.4 because of the amount of mathematical notation. Try to easy it. - Do not start a line with an equation, for instance line 100.

Relation to Prior Work: The relation to previous work is ok. However, the relation to papers [11] and [37] could be definitely discussed in more detail, as the present paper is a combination of both ideas.

Reproducibility: Yes

Additional Feedback: - When arriving to the experimental section, it is not crystal clear that the methods that the authors propose are the ones called mGAM and MAM. Can you make it more clear through the text? Especially for mGAM. - There is a lot of notation which makes some section very hard and tiring to read. What is the point of the \tau_j ? If it is of no importance it should be removed to ease notation. - Line 53: "In theory, we characterize the convergence of the proposed algorithm.". Try to be more clear with this statement. What do you mean by that? - Line 143-144: Why the sum takes all three terms in the inner optimization problem? - Line 173: I think that the authors meant \mu instead of \epsilon (maybe the epsilon notation is extracted from [11]?). Be careful, I also observed some \epsilon insetad of \mu in the supplementary material. - It is not explicit in the paper that the second set of gradients defined in line 151-152 are computed by backpropagating through the iterations of a smooth algorithm. - What is exactly the information appearing in parenthesis in Table 2 next to the averages? - In Equation 2 in appendix you also need a minimization over b otherwise if the b is made dependent of \beta the problem is non-convex.


Review 2

Summary and Contributions: This paper proposes a new class of multi-task additive models (MAM) for robust estimation and variable structure discovery, where the mode-induced metric, the structure-based regularizer, and additive hypothesis spaces are incorporated into a bilevel optimization framework. The main advantages of MAM are two-fold: MAM does not require any priori knowledge of variable structure and is robust for high-dimensional data with complex non-Gaussian noise. To implement the robust MAM efficiently, a smooth iterative algorithm is provided and its convergence is established. Empirical data experiments support the effectiveness of MAM for data-driven structure discovery and regression estimation. It is very important to investigate interpretable learning models (e.g., via data-driven structure discovery) under complex noise environment. This paper states a novel way to tackle these concerns with the help of mode-based error metric (for robust) and bilevel optimization (for interpretability).

Strengths: The data-driven automatic structure discovery has attracted increasing attention recently for interpretable machine learning , see e.g., [39][26][44][18][11]. The modal regression has been investigated for robust machine learning since the mode-induced metric is insensitive to complex noise, see, e.g.,[9][37]. To the best of my knowledge, it is novelty to explore robust data-driven structure discovery under the multi-task learning setting, e.g., the proposed multi-task additive models (MAM). In particular, the model design and bilvel optimization is significance to push the progress of structure learning for high-dimensional data. The description of the learning model is clearly and the related baseline approaches have been well summarized. The steps for the bilvel optimization have been provided in detail with theoretical foundations. Empirical evaluation for simulated and CME data demonstrate MAM’s performance (without prior structure information) for regression estimation and structure discovery in terms of multiple measures. The impacts of different noises and component functions are considered. In particular, the proposed experimental analysis provides some interpretable results for CME prediction.

Weaknesses: In theory, it may be better to provide some theoretical analysis on the generalization bounds of MAM. Though it may be challenge for learning theory due to the non-convex error metric and nonlinear additive hypothesis space (since there no related analysis even for linear mean regression [11]), the authors can provide some additional analysis on the theoretical concern. Indeed, it may be possible to characterize the generalization of MAM via multitask algorithmic stability, e.g., Zhang, Multi-Task Learning and Algorithmic Stability, AAAI, 2015. X.Wang, Junier B. Oliva, Jeff Schneider, Barnabas Poczos, Nonparametric Risk and Stability Analysis for Multi-Task Learning Problems, IJCAI, 2016. Zachary Charles, Dimitris Papailiopoulos, Stability and Generalization of Learning Algorithms that Converge to Global Optima, ICML, 2018. In addition, it may be better to provide a brief discussion for overlapping variable structure discovery [16] in the supplementary materials. %%%%%%%%%% I am satisfied with authors' response. Thus, I will keep my judgement

Correctness: I have checked the main steps of learning frameworks and optimization. Both the main claims and method are correct. The empirical analysis also is correct.

Clarity: The paper is well organized and written. However, some issues should be carefully considered to further improve the paper. The comments are given below. 1. To further improve the readability of this paper, it would be better to provide a figure to summarize the main optimization procedures for MAM. In addition, the authors also can add more notations to Table 4, e.g., key parameters involved in Section 3 and Subsection 2.4. 2. The illustration for simulated experiments (lines 196-197) should address the results of MAM does not require the prior of variable structure, while mGAM, GL, GroupSpAM depend on the given variable structure., e.g., “Even without the structure information, the proposed MAM also can provide the comparable regression estimation with mGAM (given prior structure), and usually better performance than GL, GroupSpAM. “ 3. It would be better to give more analysis about the interpretability for the result of CMEs structure discovery, although there are some illustrations in the supplementary material (lines 161-163 ). 4. The authors are advised to proofread the entire paper to further enhance the presentation of the paper. Some typos are listed as follows. (1) line 69, Page 3, “Group Lasso structure” ---->“Group Lasso”. (2) line 122, Page 5, “constructs” --> “formulates”. (3)line 165, Page 6, “were” --> “are’’. (4) line 170, Page 6, “known a prior” --> “a known prior”

Relation to Prior Work: Yes, it is. The difference between this paper and the related works has been clearly discussed in Sections 1 and 2, e.g., Figure 1, Table 1, lines 59-69, and subsections 2.2-2.3. The main contributions have been well summarized in lines 46-58.

Reproducibility: Yes

Additional Feedback:


Review 3

Summary and Contributions: This paper presents a robust additive model for multitask learning, with automatic discovery of active variables as well as variable structure in multitask regression problem. The authors use mode-induced metrics in regression to improve robustness towards complex noises, and propose a bilevel framework to screen main effect variables across all tasks and identify the group structure among variables. The proposed objective function is optimized via Prox-SAGA algorithm with theoretical convergence guarantee. Besides the theoretical analysis, the proposed model is evaluated on both synthetic and benchmark data by comparing with a wide range of related works. ================ [After reading the rebuttal] I have read the authors' feedback and other reviews. The rebuttal well addressed my concerns. I would keep my original score and recommend for acceptance.

Strengths: This work is well motivated and novel. The proposed method enjoys several nice properties: 1) a multi-task additive model that is effective for high-dimensional data with complex noises; 2) the model can consider the variable structure and sparsity without the need of any prior knowledge. Instead, the bilevel optimization enables an automatic discovery of variable structures; 3) the model is theoretically grounded. The connection with previous works is clearly analyzed and compared. The authors provide comprehensive empirical results on various datasets (synthetic and benchmark data). The proposed method is compared with a wide range of related works, via various evaluation metrics. The identified variable structure has been discussed on benchmark data (CMEs). The empirical results look promising and convincing. Moreover, the paper well written and easy to follow. The contribution is clear and well supported with theoretical and empirical results.

Weaknesses: To find the variable structure, the group number L and inactive variables |V| is needed. It would be helpful if the authors can provide some discussion on how to set these parameters. Moreover, it would be helpful to briefly discuss the advantage of using bilevel framework, i.e., in comparison with solving the inner and outer problem on a same dataset in Section 2.4.

Correctness: The proposed method is theoretically grounded. The claims look correct to me and the empirical results look comprehensive and convincing.

Clarity: The paper is clearly written and easy to follow.

Relation to Prior Work: The connection and contribution over prior works is well discussed. The proposed method is compared from both theoretical and empirical perspectives with the related works.

Reproducibility: Yes

Additional Feedback:

[Author Response · NeurIPS 2020]

**#ToR1**. **Q1: ["Weaknesses...-METHOD"].** **A1**: Thanks for your valued and constructive comments. The main motivation of our manuscript is to propose robust additive models (with theoretical analysis and applications) for realizing nonlinear estimation and structure discovery simultaneously, even data contaminated with complex noise and without priori knowledge of variable structure. There are **key differences** between our MAM and [11,37,34]: *1)Hypothesis space (nonlinear VS linear).* The proposed MAM employs the spline-based additive hypothesis space, which is more flexible than the linear assumption for regression function [11,37,34], and leads to better performance for characterizing the nonlinear relationship in data (e.g., [13,15,21,29,41]). *2)Sparsity within group ($\nu \in [0, 1]^P$ VS $\nu \equiv \mathbf{I}_P := (1, ..., 1)^T$).* The flexible selection of $\nu$ in our Outer Problem can unveil main effect variables across all tasks, which is useful to remove ambiguity for model identifiability. Indeed, [11] ignores the sparsity within group and [37,34] don't consider the structure discovery. *3) Optimization (bilevel non-convex optimization VS bilevel convex optimization).* The non-convexity of outer and inner problems is the main challenge for MAM's optimization, which can not be tackled directly by the optimization strategy for bilevel convex optimization problem[11]. A computing algorithm for bilevel non-convex scheme is proposed by developing an effective optimization method for the non-convex non-smooth inner problem. *4) Application.* As we known, MAM is the first attempt to the interpretable CMEs analysis by data-driven structure discovery strategy. In short, our work is new for structure prediction, and can't be established by easily combining [11] and [37]. We will add these remarks after Table 1 (line 69).

**Q2: ["Weaknesses..-ALGORITHM"].** **A2**: The non-convexity of outer and inner problem is the main difficulty in the optimization. To tackle this problem, an optimization strategy is proposed based on building blocks (ProxSAGA [30], HQ optimization[24] and DFBB[36]). Theorem 1 in Supp C guarantees that our computing algorithm can obtain the local optimal solution. Indeed, it's always a challenge to give the complexity analysis on bilevel optimization (we leave this direction for future work). Following your suggestion, Remark 2 (line 126 in Supp) is added to illustrate the extra complexity (computation time and space) of our optimization algorithm (compared with BiGL [11]).

**Q3: ["Correctness...-EVALUATION"].** **A3**: Thanks for your valued suggestions. Some competitors (BiGL[11], Lasso[35], RMR[37], SpAM[29]) with detailed analysis are added in Table 3 and Supp D. The AAE for each task is BiGL($11.09h$, $59.75km/s$, $63.51km/s$, $95.97km/s$, $4.77nT$), Lasso($12.16h$, $62.56km/s$, $59.81km/s$, $95.34km/s$, $4.38nT$), RMR($12.02h$, $58.65km/s$, $49.03km/s$, $98.13km/s$, $4.20nT$) and SpAM($10.02h$, $50.48km/s$, $54.97km/s$, $72.33km/s$, $3.88nT$), respectively. We can observe that MAM (in Table 3) can achieve smaller AAE than other competitors due to its ability of robust nonlinear approximation and structure discovery.

**Q4: ["Clarity-It..."].** **A4**: We have polished the manuscript following your suggestions and will proofread it carefully.

**Q5: ["Relation to .."].** **A5**: In Table 1, MAM has been related to [11,37,26,41] in terms of hypothesis space, evaluation criterion, robustness and sparsity. Additionally, we have enriched Table 1 by adding the (non)convexity of objective function, and added remark to discuss the extra complexity of MAM compared with methods[11,37].

**Q6: ["Additional feedback"].** **A6**: **[-When...]:** mGAM with an oracle variable structure is the baseline of MAM. For easy reading, we have added Remark 3 (pg.5) to state the relation of mGAM and MAM. **[-There..]:** $\tau_j$ is the weight for each group, by controlling which we can emphasize some groups that are known a priori. **[-Line 53]:** "In theory,..." -> "In theory, we provide the convergence analysis of the proposed optimization algorithm". **[-Line 143-144]:** The first term is used for robust estimation. The second term guarantees the strong convexity of transformed inner objective (See Eq.4 in Supp). The third term incorporates the variable structure $\vartheta$. **[-Line 173]:** We have corrected $\varepsilon$->$\mu$, where $\varepsilon = n\sigma^2\mu$ is the penalty parameter in the transformed inner problem (line 26 in Supp). Note that $\varepsilon$ is different from the noise $\epsilon$. **[-It...]:** A description (line 152) is added to explain that the second set of gradients is computed by backpropagating through the iterations of a smooth algorithm. **[-What...]:** The information appearing in parenthesis is standard deviation. We have revised the title of Table 2. **[-In...]:** We have revised Equation 2.

**#ToR3.** **Q1: ["Weaknesses.."].** **A1**:Thanks for the constructive comments. In Supp, Section E (pg.12) is added to discuss the generalization bound based on algorithmic stability. Firstly, we proved that the mode-induced metric satisfies Quadratic Growth condition [Charles 2018] under mild assumptions. Based on the definition of multi-task uniform stability and Theorem 2 [Zhang 2015], we quantify the stability of MAM and establish its generalization bound. In addition, a brief comparison between overlapping group lasso[16] and our MAM is provided in Supp.

**Q2: ["Clarity.."].** **A2**: 1) We have added more key parameters in Table 4 and a flowchart of optimization before starting with Section B.1. 2) The description ("Even...") is added in line 196. 3) Some competitors are added in CMEs experiment (See A3 of #ToR1). 4)We have corrected these typos and will proofread the whole manuscript.

**#ToR5.** **Q1: ["Weaknesses.."].** **A1**: A detailed description of the selection of group number $L$ and $|V|$ is added in line 171(pg.6). In fact, we have discussed the impact of group number $L$ in Fig 8, which indicates a satisfactory result can also be obtained even if $L$ is set to be larger than oracle group number. Moreover, our method can realize data-driven variable selection without the inactive variables $|V|$ being set initially (e.g., the red pixel in Figs 2 and 3).

**Q2: ["..discuss.."].** **A2**: In hyper-parameter selection problem (e.g., variable structure $\vartheta$), bilevel framework has better generalization than traditional hyper-parameter selection method (e.g, Cross-validation), since the hyper-parameter is tuned over a continuous space by minimizing the outer problem [K.P. Bennett et al., IJCNN 2006]. According to your suggestions, a further discussion on the advantage of bilevel framework is provided in Introduction (line 42).

[Meta-Review · NeurIPS 2020]

All the reviewers have supported the acceptance of this paper.